# EDGE-PRESERVING NOISE FOR DIFFUSION MODELS

## ABSTRACT

Classical generative diffusion models learn an isotropic Gaussian denoising process, treating all spatial regions uniformly, thus neglecting potentially valuable structural information in the data. Inspired by the long-established work on anisotropic diffusion in image processing, we present a novel edge-preserving diffusion model that generalizes over existing isotropic models by considering a hybrid noise scheme. In particular, we introduce an edge-aware noise scheduler that varies between edge-preserving and isotropic Gaussian noise. We show that our model's generative process converges faster to results that more closely match the target distribution. We demonstrate its capability to better learn the low-to-mid frequencies within the dataset, which plays a crucial role in representing shapes and structural information. Our edge-preserving diffusion process consistently outperforms state-of-the-art baselines in unconditional image generation. It is also more robust for generative tasks guided by a shape-based prior, such as stroke-to-image generation. We present qualitative and quantitative results showing consistent improvements (FID score) of up to 30% for both tasks. We provide source code and supplementary content via the public domain edge-preserving-diffusion.mpi-inf.mpg.de.

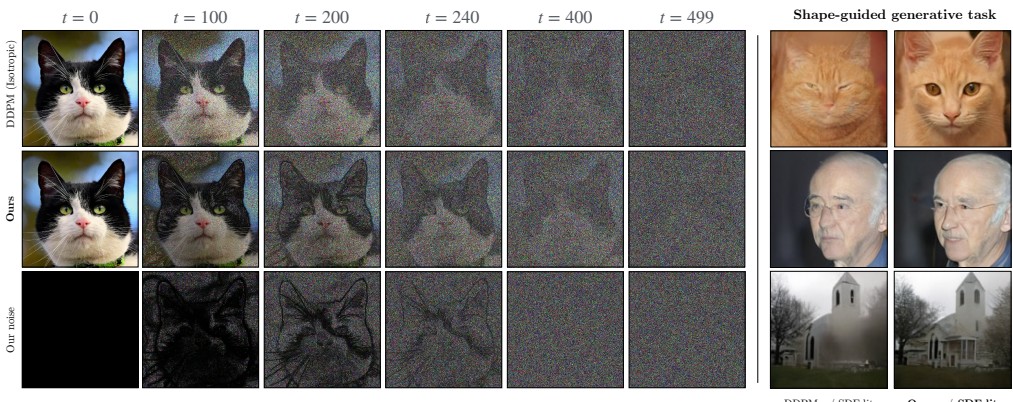

Figure 1: A classic isotropic diffusion process (top row) is compared to our hybrid edge-aware diffusion process (middle row) on the left side. We propose a hybrid noise (bottom row) that progressively changes from anisotropic ($t = 0$) to isotropic scheme ($t = 499$). We use our edge-aware noise for training and inference. On the right, we compare both noise schemes on an SDEdit framework (Meng et al., 2022) for stroke-based image generation. Our model consistently outperforms DDPM, is more robust against visual artifacts and produces sharper outputs without missing structural details.

## 1 INTRODUCTION

Previous work on diffusion models mostly uses isotropic Gaussian noise to transform an unknown data distribution into a known distribution (e.g., normal distribution), which can be analytically sampled (Song and Ermon, 2019; Song et al., 2021; Ho et al., 2020; Kingma et al., 2021). Due to the isotropic nature of the noise, all regions in the data samples $\mathbf{x}_0$ are uniformly corrupted, regardless of the underlying structural content, which is typically distributed in a non-isotropic manner. During the backward process, the model is trained to learn an isotropic *denoising* process that ignores this

potentially valuable non-isotropic information. In image processing literature (Elad et al., 2023), denoising is a well studied topic. Following the work by Perona and Malik (1990) structure-aware guidance has shown remarkable improvements in denoising. Since generative diffusion models can also be seen as *denoisers*, we ask ourselves: *Can we enhance the effectiveness of the generative diffusion process by incorporating awareness of the structural content of the data samples in the underlying dataset?*

To explore our question, we introduce a new class of diffusion models that explicitly learn a content-aware noise scheme. We call our noise scheme *edge-preserving noise*, which offers several advantages. First, it allows the backward generative process to converge more quickly to accurate predictions. Second, our edge-preserving model better captures the low-to-mid frequencies in the target dataset, which typically represent shapes and structural information. Consequently, we achieve better results for unconditional image generation. Lastly, our model also demonstrates greater robustness and quality for generative tasks that rely on shape-based priors.

To summarize, we make the following contributions:

- We present a novel class of content-aware diffusion models and show how it is a generalization of existing isotropic diffusion models

- We conduct a frequency analysis to better understand the modeling capabilities of our edge-preserving model.

- We run extensive qualitative and quantitative experiments across a variety of datasets to validate the superiority of our model over existing models.

- We observed consistent improvements in pixel space diffusion. We found that our model converges faster to more accurate predictions and better learns the low-to-mid frequencies of the target data, resulting in FID score improvements of up to 30% for unconditional image generation and most remarkably a more robust behaviour and better quality on generative tasks with a shape-based prior.

## 2 RELATED WORK

Most existing diffusion-based generative models (Sohl-Dickstein et al., 2015; Song and Ermon, 2019; Song et al., 2021; Ho et al., 2020) corrupt data samples by adding noise with the same variance to all pixels. These generative models can generate diverse novel content when the noise variance is higher. On the contrary, noise with lower variance is known to preserve the underlying content of the data samples. Rissanen et al. (2023) proposed to use an inverse heat dissipation model (IHDM), which can be seen as an isotropic Gaussian blurring model. The idea is to isotropically blur the images to corrupt them. They show that isotropic blurring in the spatial domain corresponds to adding non-isotropic noise in the frequency domain, where they run the diffusion. Hoogeboom and Salimans (2023) proposed an improved version of IHDM, where they not only blur but also use isotropic noise to corrupt data samples. The resulting model gives far better quality improvements compared to IHDM. Recently, Huang et al. (2024a) introduced correlated noise for diffusion models (BNDM). They proposed to use *blue noise* which is negatively correlated and showed noticeable improvements both in visual quality and FID scores. While IHDM and BNDM also consider a form of non-isotropic noise, they do not explicitly take into account the structures present in the signal.

Various efforts (Bansal et al., 2023; Daras et al., 2023) were made to develop non-isotropic noise models for diffusion processes. Dockhorn et al. (2022) proposed to use critically-damped Langevin diffusion where the data variable at any time is augmented with an additional "velocity" variable. Noise is only injected in the velocity variable. Voleti et al. (2022) performed a limited study on the impact of isotropic vs non-isotropic Gaussian noise for a score-based model. The idea behind non-isotropic Gaussian noise is to use noise with different variance across image pixels. They use a non-diagonal covariance matrix to generate non-isotropic Gaussian noise, but their sample quality did not improve in comparison to the isotropic case. Yu et al. (2024) developed this idea further and proposed a Gaussian noise model that adds noise with non-isotropic variance to pixels. The variance is chosen based on how much a pixel or region needs to be edited. They demonstrated a positive impact on editing tasks.

Our definition of anisotropy follows directly from the seminal work by Perona and Malik (1990) on anisotropic diffusion for image filtering. We apply a non-isotropic variance to pixels in an edge-aware manner, meaning that we suppress noise on edges.

# 3 BACKGROUND

**Generative diffusion processes.** A generative diffusion model consists of two processes: the forward process transforms data samples $\mathbf{x}_0$ into samples $\mathbf{x}_T$ that are distributed according to a well-known prior distribution, such as a normal distribution $\mathcal{N}(0, I)$. The corresponding backward process does exactly the opposite: it transforms samples $\mathbf{x}_T$ into $\hat{\mathbf{x}}_0$, distributed according to the target distribution $p_0(\mathbf{x})$. This backward process involves predicting a vector quantity, interpretable as either noise or the gradient of the data distribution, which is precisely the task for which the generative diffusion model is trained. Previous works (Song and Ermon, 2019; Song et al., 2021; Ho et al., 2020; Kingma et al., 2021; Rissanen et al., 2023; Hoogeboom and Salimans, 2023) typically formulate the forward process as the following linear equation:

$$\mathbf{x}_t = \gamma_t \mathbf{x}_0 + \sigma_t \boldsymbol{\epsilon}_t \tag{1}$$

here, $\mathbf{x}_t$ is the data sample diffused up to time $t$, $\mathbf{x}_0$ stands for the original data sample, $\boldsymbol{\epsilon}_t$ is a standard normal Gaussian noise, and the *signal coefficient* $\gamma_t$ and *noise coefficient* $\sigma_t$ determine the signal-to-noise ratio (SNR) ($\gamma_t/\sigma_t$). The SNR refers to the proportion of signal retained relative to the amount of injected noise. Note that $\gamma_t$ and $\sigma_t$ are both scalars. Previous works have made several different choices for $\gamma_t$ and $\sigma_t$ respectively, leading to different variants, each with their own advantages and limitations. Typically, a *noise schedule* $\beta_t$ is employed to govern the rate at which $\gamma_t$ and $\sigma_t$ vary over time (Ho et al., 2020). Let us first define $\alpha_t = 1 - \beta_t$ and $\bar{\alpha}_t = \prod_{i=1}^{t} \alpha_i$. Song and Ermon (2019) then defined their forward diffusion process with $\gamma_t = 1$ and $\sigma_t = \sqrt{1 - \bar{\alpha}_t}$, leading to a process with exploding variance. Instead, Ho et al. (2020) chose $\gamma_t = \sqrt{\bar{\alpha}_t}$ and $\sigma_t = \sqrt{1 - \bar{\alpha}_t}$ which better preserves variance.

**Diffusion model paradigms.** There are two widely-adopted general mathematical frameworks that form the major "paradigms" of generative diffusion models. The first paradigm is *score-based generative modelling*, introduced by Song and Ermon (2019). In this paradigm, the generative model essentially learns to estimate the gradients of the target distribution, allowing for iterative sampling via these gradients. A prominent type of diffusion model that is based on this paradigm are *SDE-based diffusion models*, which simulate the generative backward process as a stochastic differential equation to take samples. In Appendix A.1, we perform a theoretical analysis showing how our edge-preserving diffusion process can be interpreted under this paradigm.

*Denoising probabilistic modeling* paradigm, introduced by Ho et al. (2020) models the generative process as a well-studied Gaussian process with known posteriors. The reverse process is then modelled as a parameterized Markov chain that approximates this Gaussian process. In this paper, we adopt this paradigm as it aligns well with the statistical property we are investigating: non-isotropic variance driven by image content. Flow matching (Lipman et al., 2022) is another class of generative models that are gaining a lot of traction. In Appendix B, we discuss in detail how our framework can be adapted to flow matching.

**Denoising probabilistic model.** Following the probabilistic paradigm, we would like to introduce the posterior probability distributions of the general diffusion process described by Eq. (1). We will show the exact form that our forward and backward processes take in Section 4.1 and Section 4.3 respectively. For details and full derivations of the equations in this section, we would like to refer to the appendix of Kingma et al. (2021). The isotropic diffusion process formulated in Eq. (1) has the following marginal distribution:

$$q(\mathbf{x}_t|\mathbf{x}_0) = \mathcal{N}(\gamma \mathbf{x}_0, \sigma_t^2 \boldsymbol{I}) \tag{2}$$

Moreover, it has the following Markovian transition probabilities:

$$q(\mathbf{x}_t|\mathbf{x}_s) = \mathcal{N}(\gamma_{t|s}\mathbf{x}_s, \sigma_{t|s}^2 \boldsymbol{I}) \tag{3}$$

with the forward posterior signal coefficient $\gamma_{t|s} = \frac{\gamma_t}{\gamma_s}$ and the forward posterior variance (or square of the noise coefficient) $\sigma_{t|s}^2 = \sigma_t^2 - \gamma_{t|s}^2 \sigma_s^2$ and $0 < s < t < T$. For a Gaussian diffusion process,

given that $q(\mathbf{x}_s|\mathbf{x}_t, \mathbf{x}_0) \propto q(\mathbf{x}_t|\mathbf{x}_s)q(\mathbf{x}_s|\mathbf{x}_0)$, one can analytically derive a *backward process* that is also Gaussian, and has the following marginal distribution:

$$q(\mathbf{x}_s|\mathbf{x}_t, \mathbf{x}_0) = \mathcal{N}(\boldsymbol{\mu}_{t \to s}, \sigma_{t \to s}^2 \boldsymbol{I}). \tag{4}$$

The backward posterior variance $\sigma_{t \to s}^2$ has the following form:

$$\sigma_{t \to s}^2 = \left( \frac{1}{\sigma_s^2} + \frac{\gamma_{t|s}^2}{\sigma_{t|s}^2} \right)^{-1} \tag{5}$$

and the backward posterior mean $\boldsymbol{\mu}_{t \to s}$ is formulated as:

$$\boldsymbol{\mu}_{t \to s} = \sigma_{t \to s}^2 \left( \frac{\gamma_{t|s}}{\sigma_{t|s}^2} \mathbf{x}_t + \frac{\gamma_s}{\sigma_s^2} \mathbf{x}_0 \right). \tag{6}$$

Samples can be generated by simulating the reverse Gaussian process with the posteriors in Eq. (5) and Eq. (6). A practical issue is that Eq. (6) itself depends on the unknown $\mathbf{x}_0$, the sample we are trying to generate. To overcome this, one can instead approximate the analytic reverse process in which $\mathbf{x}_0$ is replaced by its approximator $\hat{\mathbf{x}}_0$, learned by a deep neural network $f_\theta(\boldsymbol{x}_t, t)$. The network can learn to directly predict $\mathbf{x}_0$ given an $\mathbf{x}_t$ (a sample with a level of noise that corresponds to time $t$), but previous work has shown that it is beneficial to instead optimize the network to learn the approximator $\hat{\boldsymbol{\epsilon}}_t$. $\hat{\boldsymbol{\epsilon}}_t$ predicts the unscaled Gaussian white noise that was injected at time $t$. $\hat{\mathbf{x}}_0$ can then be obtained via Eq. (7), which follows from Eq. (1).

$$\hat{\mathbf{x}}_0 = \frac{1}{\gamma_t} \mathbf{x}_t - \frac{\sigma_t}{\gamma_t} \hat{\boldsymbol{\epsilon}}_t \tag{7}$$

**Edge-preserved filtering in image processing.**   In this work, we aim to choose $\gamma_t$ and $\sigma_t$ such that we obtain a diffusion process that injects noise in a content-aware manner. To do this, we are inspired by the field of image processing, where a classic and effective technique for denoising is edge-preserved filtering via *anisotropic diffusion* (Weickert, 1998). Perona and Malik (1990) showed that a Gaussian scale-space (a family of images obtained by convolving the original image $\mathbf{x}_0$ with Gaussian kernels of varying scales) can also be expressed in terms of a diffusion process. However, if the goal is to remove noise from an image, a disadvantage of a standard isotropic blurring process is that the relevant structural information in the image also gets distorted. To overcome this problem, Perona and Malik (1990) propose an anisotropic diffusion process of the form:

$$\mathbf{x}_t = \mathbf{x}_0 + \int_0^t \mathbf{c}(\mathbf{x}_s, s) \Delta \mathbf{x}_s \, ds \tag{8}$$

where the diffusion coefficient $\mathbf{c}(\mathbf{x}_s, s)$ takes the following form:

$$\mathbf{c}(\mathbf{x}, t) = \frac{1}{\sqrt{1 + \frac{||\nabla \mathbf{x}_t||}{\lambda}}} \tag{9}$$

where $||\nabla \mathbf{x}||$ is the gradient magnitude image, and $\lambda$ is the *edge sensitivity*. Intuitively, in the regions of the image where the gradient response is high (on edges), the diffusion coefficient will be smaller, and therefore the signal gets less distorted there. The edge sensitivity $\lambda$ determines how sensitive the diffusion coefficient is to the image gradient response.

Inspired by the anisotropic diffusion coefficient presented in Eq. (9), we aim to design a *linear diffusion process* that incorporates edge-preserving noise. Our hope is that by doing this, the generative diffusion model will better learn the underlying structures of the target distribution, leading to a more effective generative denoising process. To obtain our content-aware linear diffusion process, we apply the idea of edge-preserved filtering to the noise term of Eq. (1). However, it is important to note that we cannot directly adopt the formulation of (Perona and Malik, 1990). In the original formulation, the diffusion coefficient $\mathbf{c}(\mathbf{x}, t)$ is time-dependent (it depends on the state of the data sample $\mathbf{x}_t$ at each point in time) and therefore the resulting forward process would no longer be linear. To overcome this, we let our diffusion coefficient only depend on $\mathbf{x}_0$. Intuitively, this preserves edges by suppressing noise based on the *original* image content. When edge preservation is applied to blurring (such as in Perona and Malik (1990)), this workaround fails because the edges gradually

"blend" into each other over time, making the dependency on $\mathbf{x}_t$ essential. Instead, we operate on the noise, and define an edge-preserving forward process as follows:

$$\mathbf{x}_t = \gamma_t \mathbf{x}_0 + \frac{b}{\sqrt{1 + \frac{||\nabla \mathbf{x}_0||}{\lambda(t)}}} \boldsymbol{\epsilon}_t \tag{10}$$

Where $b$ is the noise coefficient's numerator and can be chosen as desired. To study the impact of non-isotropic edge-preserving noise on the generative diffusion process, we choose our parameters $\gamma_t = \sqrt{\bar{\alpha}_t}$ and $b = \sqrt{1 - \bar{\alpha}_t}$ such that it closely matches the well-studied forward process of (Ho et al., 2020), but nothing prevents us from making different choices for $\gamma_t$ and $b$. Note that the noise coefficient in Eq. (1) becomes a tensor $\boldsymbol{\sigma}_t$ instead of a scalar $\sigma_t$ for our process. Intuitively, we suppress noise on the edges and leave noise unchanged in more uniform image regions. In our formulation, we also consider $\lambda$ to be time-varying (more details in section Section 4.2).

## 4  AN EDGE-PRESERVING GENERATIVE PROCESS

### 4.1  FORWARD HYBRID NOISE SCHEME

The forward *edge-preserving* process described in Eq. (10) in its pure form is not very meaningful in our setup. This is because if the edges are preserved all the way up to time $t = T$, we end up with a rather complex distribution $p_T(x)$ that we cannot analytically take samples from. Instead, we would like to end up with a well-known *prior* distribution at time $t = T$, such as the standard normal distribution. To achieve this, we instead consider the following hybrid forward process:

$$\mathbf{x}_t = \gamma_t \mathbf{x}_0 + \frac{b}{(1 - \tau(t))\sqrt{1 + \frac{||\nabla \mathbf{x}_0||}{\lambda(t)}} + \tau(t)} \boldsymbol{\epsilon}_t \tag{11}$$

A more general form of this equation can be found in Appendix A.1. The function $\tau(t)$ now appearing in the denominator of the diffusion coefficient is the *transition function*. When $\tau(t) < 1$, we obtain edge-preserving noise (the edge-preservation is stronger when $\tau(t) \approx 0$). The turning point where $\tau(t) = 1$ is called the *transition point* $t_\Phi$. At the transition point, we switch over to isotropic noise with scalar noise coefficient $\sigma_t = b$ (note that we chose $\gamma_t = \sqrt{\bar{\alpha}_t}$ and $b = \sqrt{1 - \bar{\alpha}_t}$). This approach allows us to flexibly design noise schedulers that start off with edge-preserving noise and towards the end of the forward process fall back to an isotropic diffusion coefficient. Practically, one can choose any function for $\tau(t)$, as long as it maps to $[0; 1]$ and $\tau(t) = 1$ for $t$ in proximity to $T$. We performed an ablation for different transition functions in Section 5.2. Observe how our diffusion process generalizes over existing isotropic processes: by setting $\tau(t) = 1$ constant, we simply obtain an isotropic process with signal coefficient $\gamma_t$ and noise coefficient $\sigma_t = b$. Choosing any other non-constant function for $\tau(t)$ leads to a hybrid diffusion process that consists of a mix of an edge-preserving stage and an isotropic stage (at $\tau(t) = 1$).

### 4.2  TIME-VARYING EDGE SENSITIVITY $\lambda(t)$

The edge sensitivity parameter $\lambda$ controls the granularity of the content preservation on image edges. By setting $\lambda$ very low, all levels of edges and fine details are preserved. The higher we increase $\lambda$, the less details will be preserved. By choosing a very high $\lambda$ (e.g. $\lambda > 1e^{-1}$), the edge-preserving stage of the diffusion process almost resembles an isotropic process because the noise suppression on the edges is unnoticeable. We study this impact in detail in the ablation study in Section 5.2.

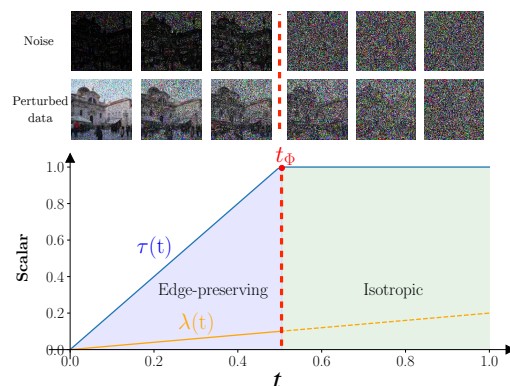

We found that choosing constant values for $\lambda$ has negative impact on sample quality. Selecting a $\lambda$-value that is too low results in unrealistic, "cartoonish" outputs, while a $\lambda$-value that is too high diminishes the effectiveness of the edge-preserving diffusion model, making it nearly indistinguishable from an isotropic model.

Figure 2: We visually compare the impact of our anisotropic edge-preserving noise on the generation (reverse) process. In each column, we show predictions $\hat{x}_0$ at selected time steps. Our method converges significantly faster to a sharper and less noisy image than its isotropic counterpart. This is evident by the earlier emergence (from $t = 400$) of structural details like the pattern on the cat's head, eyes, and whiskers with our approach.

To overcome this, we instead consider a time-varying edge sensitivity $\lambda(t)$. We set an interval $[\lambda_{min}; \lambda_{max}]$ that bounds the possible values for the time-varying edge sensitivity. The function that governs $\lambda(t)$ within this interval can in theory again be chosen freely. We have so far experimented with a linear function and a sigmoid function. We experienced that a linear function for $\lambda(t)$ resulted in higher sample quality and therefore used this function for our experiments. Additionally, we have attempted to optimize the interval $[\lambda_{min}; \lambda_{max}]$, but this led to unstable behaviour.

## 4.3 BACKWARD PROCESS POSTERIORS AND TRAINING

Given our forward hybrid diffusion process introduced in Section 4.1, we can derive the actual formulations for the posterior mean $\boldsymbol{\mu}_{t \to s}$ and variance $\boldsymbol{\sigma}^2_{t \to s}$ for the corresponding backward process. To do this, we simply fill in Eq. (5) and Eq. (6) with our choices for the signal coefficient $\gamma_t$ and variance $\boldsymbol{\sigma}_t^2$. Recall that we chose $\boldsymbol{\sigma}_t^2$ to be a tensor, which is why the backward posterior variance $\boldsymbol{\sigma}^2_{t \to s}$ is again a tensor, contrary to isotropic diffusion processes considered in previous works. Regardless, we can use the same equations and the algebra still works.

We first introduce an auxiliary variable $\boldsymbol{\sigma}^2(t)$, which represents the variance of our forward process at a given time $t$. This is simply the square of our choice for the noise coefficient $\boldsymbol{\sigma}_t$ formulated in Eq. (11):

$$\boldsymbol{\sigma}^2(t) = \frac{1 - \bar{\alpha}_t}{(1 - \tau(t))^2 \left(1 + \frac{||\nabla \mathbf{x}_0||}{\lambda(t)}\right) + 2\left((1 - \tau(t))\sqrt{1 + \frac{||\nabla \mathbf{x}_0||}{\lambda(t)}}\tau(t)\right) + \tau(t)^2} \quad (12)$$

Here $\bar{\alpha}_t$ has the same meaning as earlier described in Section 3. We now have the backward posterior variance $\boldsymbol{\sigma}^2_{t \to s}$:

$$\boldsymbol{\sigma}^2_{t \to s} = \left(\frac{1}{\boldsymbol{\sigma}^2(t)} + \frac{\frac{\bar{\alpha}_t}{\bar{\alpha}_s}}{\boldsymbol{\sigma}^2(t) - \frac{\bar{\alpha}_t}{\bar{\alpha}_s}\boldsymbol{\sigma}^2(s)}\right)^{-1} \quad (13)$$

and the backward posterior mean $\boldsymbol{\mu}_{t \to s}$:

$$\boldsymbol{\mu}_{t \to s} = \boldsymbol{\sigma}^2_{t \to s}\left(\frac{\frac{\sqrt{\bar{\alpha}_t}}{\sqrt{\bar{\alpha}_s}}}{\boldsymbol{\sigma}^2(t) - \frac{\bar{\alpha}_t}{\bar{\alpha}_s}\boldsymbol{\sigma}^2(s)}\mathbf{x}_t + \frac{\sqrt{\bar{\alpha}_s}}{\boldsymbol{\sigma}^2(s)}\mathbf{x}_0\right) \quad (14)$$

Given Eq. (13) and Eq. (14), the only unknown preventing us from simulating the Gaussian backward process is $\mathbf{x}_0$.

Note that $\mathbf{x}_0$ in our case depends on a non-isotropic noise. Therefore, we cannot just use an isotropic approximator $\hat{\boldsymbol{\epsilon}}_t$ for the isotropic noise $\boldsymbol{\epsilon}_t$ to predict $\hat{\mathbf{x}}_0$ via Eq. (7). Instead, we need a model that can predict the non-isotropic noise $\boldsymbol{\sigma}_t\boldsymbol{\epsilon}_t$ . We introduce the loss function that trains such an approximator:

$$\mathcal{L} = ||f_\theta(\boldsymbol{x}_t, t) - \boldsymbol{\sigma}_t\boldsymbol{\epsilon}_t||^2. \quad (15)$$

It is very similar to the loss function used in DDPM, **with the difference that our model explicitly learns to predict the non-isotropic ($\boldsymbol{\sigma}_t\boldsymbol{\epsilon}_t$) edge-preserving noise**. In Appendix D, we show how

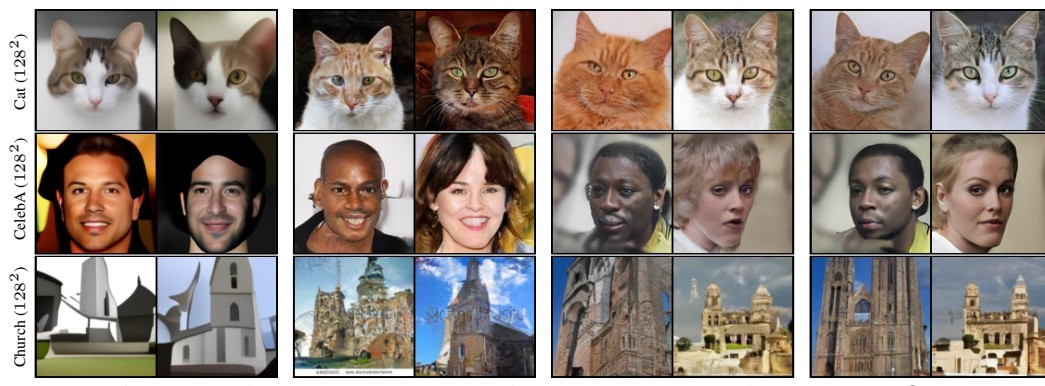

IHDM (Rissanen et al.)    BNDM (Huang et al.)    DDPM (Ho et al.)    **Ours**

Figure 3: We compare unconditionally generated samples for IHDM, BNDM, DDPM with our model. Ours perform consistently better both qualitatively and quantitatively. Corresponding FID scores can be found in Table 1. More results are presented in the appendix.

our formulation can be adapted to approximate the negative log-likelihood. $f_\theta(\boldsymbol{x}_t, t)$ stands for the time-conditioned U-Net used to approximate the time-varying noise function. The visual difference between the backward process of an isotropic diffusion model (DDPM) and ours is shown in Fig. 2. Our formulation introduces a negligible overhead. The only additional computation that needs to be performed is the image gradient $||\nabla \mathbf{x}||$, which can very efficiently be done on modern GPUs. We have not noticed any significant difference in training times between vanilla DDPM and our method.

## 5 EXPERIMENTS

**Implementation details**    We compare our method against four baselines, namely DDPM (Ho et al., 2020), Simple Diffusion (Hoogeboom et al., 2023), IHDM (Rissanen et al., 2023) and BNDM (Huang et al., 2024a). The motivation for comparing with the latter two works is that they also consider a non-isotropic form of noise. In Appendix B, we discuss how our method could be extended to continuous normalizing flows via Flow Matching (Lipman et al., 2022).

We perform experiments on two settings: pixel-space diffusion following the setting of Ho et al. (2020); Rissanen et al. (2023) and latent-space diffusion following (Rombach et al., 2022) noted as LDM in Table 1, where the diffusion process can be driven by any method but runs in the latent space. We use the following datasets: CelebA ($128^2$, 30,000 training images) (Lee et al., 2020), AFHQ-Cat ($128^2$, 5,153 training images) (Choi et al., 2020), Human-Sketch ($128^2$, 20,000 training images) (Eitz et al., 2012) (see Appendix) and LSUN-Church ($128^2$, 126,227 training images) (Yu et al., 2015) for pixel-space diffusion. For high-resolution image generation with latent-space diffusion (Rombach et al., 2022), we use on CelebA ($256^2$), AFHQ-Cat ($512^2$).

We used a batch size of 64 for all experiments in image space, and a batch size of 128 for all experiments in latent space. We trained AFHQ-Cat ($128^2$) for 1000 epochs, AFHQ-Cat ($512^2$) (latent diffusion) for 1750 epochs, CelebA($128^2$) for 475 epochs, CelebA($256^2$) (latent diffusion) for 1000 epochs and LSUN-Church($128^2$) for 90 epochs for our method and all baselines we compare to. Our framework is implemented in Pytorch (Paszke et al., 2017). For the network architecture we use a 2D U-Net from Rissanen et al. (2023). We use T = 500 discrete time steps for both training and inference, except for AFHQ-Cat128, where we used T = 750. To optimize the network parameters, we use Adam optimizer (Kingma and Ba, 2014) with learning rate $1e^{-4}$ for latent-space diffusion models and $2e^{-5}$ for pixel-space diffusion models. We trained all datasets on 2x NVIDIA Tesla A40.

For our final results in image space, we used a linear scheme for $\lambda(t)$ that linearly interpolates between $\lambda_{min} = 1e^{-4}$ and $\lambda_{max} = 1e^{-1}$. We used a transition point $t_\Phi = 0.5$ and a linear transition function $\tau(t)$. For latent diffusion, we used $\lambda_{min} = 1e^{-5}$ and $\lambda_{max} = 1e^{-1}$, with $t_\Phi = 0.5$ and a linear $\tau(t)$.

To evaluate the quality of generated samples, we consider FID (Heusel et al., 2017). using the implementation from Stein et al. (2024), with Inception-v3 network (Szegedy et al., 2016) as

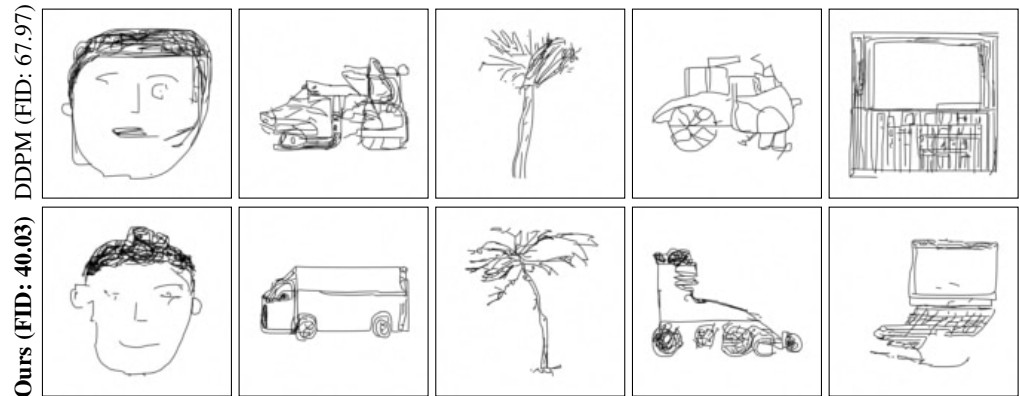

Figure 4: Unconditional samples for the Human Sketch ($128^2$) dataset (Eitz et al., 2012). All models were trained for an equal amount of 575 epochs. Our edge-aware model outperforms all models including DDPM (see Fig. 6).

backbone. We generate 30k images to compute FID scores for unconditional generation for all datasets.

**Unconditional image generation**  We show unconditional image generation results in Fig. 3 and Appendix E. The corresponding FID metrics are listed in Table 1. We observe improvements w.r.t. all baselines both visually and quantitatively. While the visual improvement over DDPM is subtle, our model generally demonstrated greater robustness to artifacts. We attribute these improvements to the explicit training of our model on predicting the non-isotropic noise associated with the edges in the dataset. We also performed comparisons in the latent space, which are listed in Table 3 in Appendix E. For latent space diffusion (CelebA($256^2$) and AFHQ-Cat($512^2$)), although our model is slightly outperformed on the FID metric, the visual quality of our samples is often comparable, and at times even superior (see Fig. 12 and Fig. 13 in Appendix E). This highlights the known limitations of FID, as it doesn't always reliably capture visual quality (Liu et al., 2018). In particular we would also like to draw attention to Fig. 4. The Human-Sketch ($128^2$) dataset (Eitz et al., 2012) is very uncommon to use for evaluation in the diffusion community, however we found it interesting given its content is entirely composed of edges. Results for additional baselines are shown in Appendix E. Training and inference time and memory consumptions of all methods are shown in Table 6.

Table 1: Quantitative FID score comparisons among IHDM (Rissanen et al., 2023), DDPM (Ho et al., 2020), BNDM (Huang et al., 2024a) and our method across different datasets.

| FID ($\downarrow$) | CelebA($128^2$) | LSUN-Church($128^2$) | AFHQ-Cat($128^2$) |
|---|---|---|---|
| IHDM | 89.67 | 119.34 | 53.86 |
| DDPM | 28.17 | 31.00 | 17.60 |
| BNDM | 26.35 | 29.86 | 14.54 |
| Ours | **26.15** | **23.17** | **13.06** |

**Stroke-guided image generation (SDEdit)**  Motivated by the hope that our model would better adhere to the guidance provided by shape-based priors such as sketches, we applied our edge-preserving diffusion model to the SDEdit framework  (Meng et al., 2022) for sketch-guided image generation. More specifically, we converted a set of 1000 original images from the training set into stroke painting images using k-means clustering. To generate samples, we then use the stroke paintings as inputs to the SDEdit framework, for different diffusion models including BNDM (Huang et al., 2024a), DDPM (Ho et al., 2020), Simple Diffusion (SimpleDiff) (Hoogeboom et al., 2023). We used a hijack point of $0.55T$, meaning that we run 55% of the forward process to get a noisy stroke painting, which is then denoised to obtain the sample. We computed the FID score between the set of original images and the set of generated samples, to measure which method is able to make better reconstructions given a shape-based prior. A comparison is shown in Fig. 5. We observed that our edge-preserving model better adheres to the guiding prior and overall behaves more robustly on this task. It suffers less from artifacts that are present for the other methods, leading to a significant

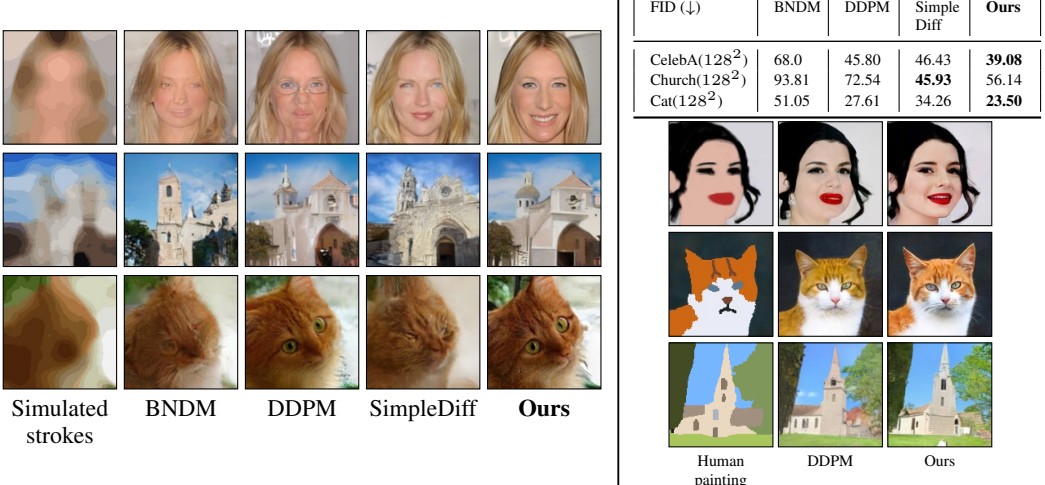

| FID (↓) | BNDM | DDPM | Simple Diff | **Ours** |
|---|---|---|---|---|
| CelebA($128^2$) | 68.0 | 45.80 | 46.43 | **39.08** |
| Church($128^2$) | 93.81 | 72.54 | **45.93** | 56.14 |
| Cat($128^2$) | 51.05 | 27.61 | 34.26 | **23.50** |

Simulated strokes | BNDM | DDPM | SimpleDiff | **Ours**

Human painting | DDPM | Ours

Figure 5: **Left:** Various diffusion models applied to the SDEdit framework (Meng et al., 2022) are shown. The leftmost column displays the stroke-based guide, with the other three columns showing the model outputs. Overall our model shows sharper details with less distortions compared to other models, leading to a better visual and quantitative performance. The corresponding FID scores are shown in the top right column. **Right:** Our model also effectively uses human-drawn paintings as shape guides, with particularly precise adherence to details, such as the orange patches on the cat's fur, unlike DDPM (middle column).

improvement in performance both qualitatively and quantitatively. We evaluate precision/recall metrics (Kynkäänniemi et al., 2019) in Tables 7 and 8 to show our method does not limit diversity. We also compare CLIP score (Radford et al., 2021) in Table 10, showing our model better preserves the semantics than Simple Diffusion and DDPM. Our edge-preserving framework can be seamlessly integrated into existing diffusion-based algorithms like RePaint (Lugmayr et al., 2022) for image inpainting, as shown in Table 9 and Figure 14.

## 5.1 FREQUENCY ANALYSIS OF TRAINING PERFORMANCE

To better understand our model's capacity of modeling the target distribution, we conducted an analysis on its training performance for different frequency bands. Our setup is as follows, we create 5 versions of the AFHQ-Cat128 dataset, each with a different cutoff frequency. This corresponds to convoluting each image in the dataset with a Gaussian kernel of a specific standard deviation $\sigma$, representing a frequency band. For each frequency band, we then trained our model for a fixed amount of 10000 training iterations. We place a model checkpoint at every 1000 iterations, so we can also investigate the evolution of the performance over this training time. We measure the performance by computing the FID score between 1000 generated samples (for that specific checkpoint) and the original dataset of the corresponding frequency band. A visualization of the analyzed results is presented in the inline figure on the right. We found that our model

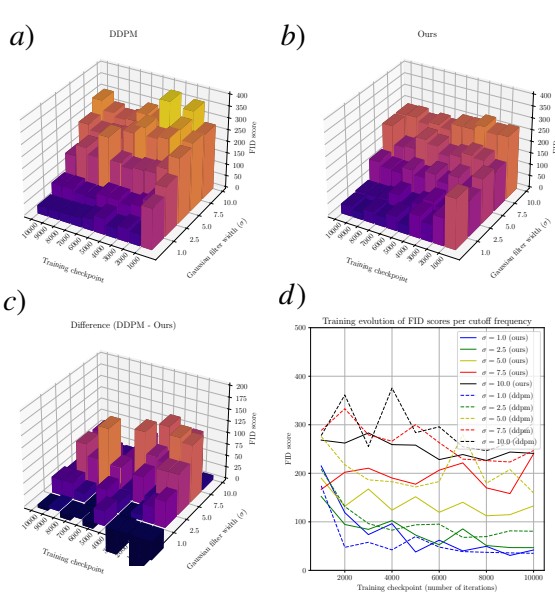

is able to learn the low-to-mid frequencies of the dataset significantly better than the isotropic model (DDPM). The figure shows the evolution of FID score over the first 10,000 training iterations per frequency band (larger $\sigma$ values correspond to lower frequency bands). $a)$ and $b)$ show performance in terms of FID score of DDPM and our model, respectively. $c)$ shows their difference (positive favors our method). $d)$ visualizes the information in 2D for a more accurate comparison. Our model significantly outperforms in low-to-mid frequency bands (lower FID is better).

## 5.2 ABLATION STUDY

**Impact of transition function** $\tau(t)$**.** We have experimented with three different choices for the transition function $\tau(t)$: linear, cosine and sigmoid. While cosine and sigmoid show similar performance, we found that having a smooth linear transition function significantly improves the performance of the model. A qualititative and quantitative comparison between the choices is presented in the inline figure below.

**Impact of transition points** $t_\Phi$**.** We have investigated the impact of the transition point $t_\Phi$ on our method's performance by considering 3 different diffusion schemes: 25% edge-preserving - 75% isotropic, 50% isotropic - 50% edge-preserving and 75% edge-preserving - 25% isotropic. A visual example for AFHQ-Cat ($128^2$) is presented in the inline figure on the right. We have experienced that there are limits to how far the transition point can be placed without sacrificing sample quality. Visually, we observe that the further the transition point is placed, the less details the model generates. The core shapes however stay intact. This is illustrated well by Fig. 7 in Appendix E. For the datasets we tested on, we found that the 50%-50% diffusion scheme works best in terms of FID metric and visual sharpness. This again becomes apparent in Fig. 7: although the samples for $t_\Phi = 0.25$ contain slightly more details, the samples for $t_\Phi = 0.5$ are significantly sharper.

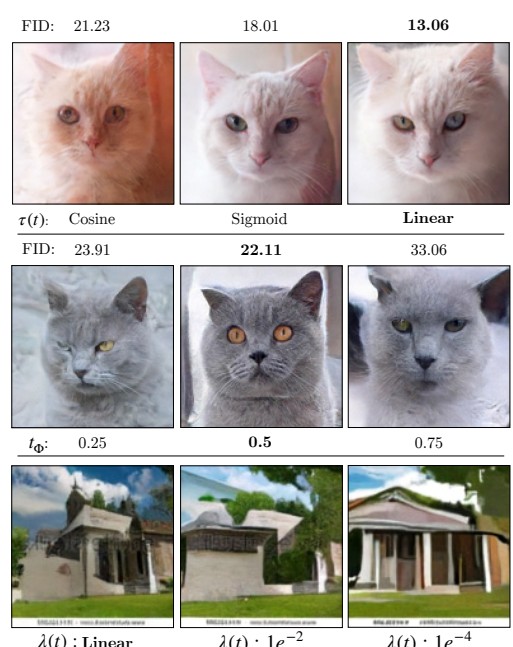

**Impact of edge sensitivity** $\lambda(t)$**.** As shown in the above inline figure, lower constant $\lambda(t)$ values lead to less detailed, more flat, "water-painting-style" results. The intuition behind this is that a lower $\lambda(t)$ corresponds to stronger edge-preserving noise and our model is explicitly trained to accordingly better learn the core structural shapes instead of the high-frequency details that we typically find in interior regions. Our time-varying choice for $\lambda(t)$ works better than other settings in our experiments, by effectively balancing the preservation of structural information across different granularities.

## 6 CONCLUSION

We introduced a new class of edge-preserving generative diffusion models that extend existing isotropic models with negligible overhead. In practice, we didn't notice any increase in training/inference time. Our linear diffusion process operates in two stages: an edge-preserving phase followed by an isotropic phase. The edge-preserving stage maintains core shapes, while the isotropic stage fills in details. This decoupled approach captures low-to-mid frequencies better and converges to sharper predictions faster. It improves performance on both unconditional and shape-guided tasks, outperforming several state-of-the-art models. For future work it would be interesting to extend our non-isotropic framework to the temporal dimension for video generation to improve time-consistency of important image features.

**Ethics statement.** We recognize that releasing generative models can open doors for malevolent actors to exploit and make misuse of them. However, by being transparent about the workings of our method and releasing the source code, we hope to support the research communities that are working on methods to better detect machine-generated content and warn or protect against it.

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

# A  RELATION TO SCORE-BASED GENERATIVE MODELING

## A.1  TRAINING OF A SCORE-BASED MODEL

Given *any* $\mathbb{R}^d$-valued ($d \in \mathbb{N}$) forward process $(\mathbf{x}_t)_{t \in [0, T]}$ such that $\mathbf{x}_0$ is distributed to a desired data distribution $\mu$ on $\mathbb{R}^d$, a score-based model can be trained by minimizing the loss:

$$\mathcal{L}(\tilde{s}) := \int_0^T \alpha(t) \int \mu(\mathrm{d}x) \, \mathrm{E}_x \left[ \left\| \tilde{s}(t, \mathbf{x}_t) \right\|^2 + 2 \nabla_x \cdot \tilde{s}(t, \mathbf{x}_t) \right] \mathrm{d}t, \tag{16}$$

where $T \in (0, \infty)$, $\alpha : [0, T] \to [0, \infty)$ is a suitable *weighting* function and $\tilde{s} : [0, T] \times \mathbb{R}^d \to \mathbb{R}^d$ is the desired score estimate. The *score* is defined to be

$$s(t, \, \cdot \, ) := \nabla \ln p_t, \tag{17}$$

where $p_t$ denotes the density of $\mathbf{x}_t$ with respect to the Lebesgue measure on $\mathbb{R}^d$, which we assume to exist for all $t \in [0, T]$.

In order to ensure stability and convergence of the training, $\alpha(t)$ is usually chosen to be inversely proportional to the expected squared norm:

$$\mathrm{E} \left[ \left\| s(t, \mathbf{x}_t) \right\|^2 \right] \tag{18}$$

of the true score $s(t, \, \cdot \, )$.

In practice, $\mathbf{x}_t$ is often conditionally Gaussian given $\mathbf{x}_0$. In that case, the suggested choice for $\alpha(t)$ can be easily computed. In fact, the score of a Gaussian random variable with covariance matrix $\Sigma$ is given by:

$$\mathrm{tr} \left( \Sigma^{-1} \right). \tag{19}$$

## A.2  SAMPLING IN A SCORE-BASED MODEL

Assuming that $(\mathbf{x}_t)_{t \in [0, T]}$ is the solution of a stochastic differential equation (SDE)

$$\mathrm{d}\mathbf{x}_t = b(t, \mathbf{x}_t) \, \mathrm{d}t + \sigma(t, \mathbf{x}_t) \, \mathrm{d}\mathbf{w}_t \tag{20}$$

for some *drift* $b : [0, T] \times \mathbb{R}^d \to \mathbb{R}^d$, *diffusion coefficient* $\sigma : [0, T] \times \mathbb{R}^d \to \mathbb{R}^{d \times d}$ and Wiener process $(\mathbf{w}_t)_{t \in [0, T]}$, a mild condition (Haussmann and Pardoux, 1986) on the drift and diffusion coeffcient are sufficient to show that the *reverse* process

$$\overline{\mathbf{x}}_t := \mathbf{x}_{T-t} \quad \text{for } t \in [0, T] \tag{21}$$

is the solution of an SDE as well. In fact, in that case, $(\overline{\mathbf{x}}_t)_{t \in [0, T]}$ is the solution

$$\mathrm{d}\overline{\mathbf{x}}_t = \overline{b}(t, \overline{\mathbf{x}}_t) \, \mathrm{d}t + \overline{\sigma}(t, \overline{\mathbf{x}}_t) \, \mathrm{d}\overline{\mathbf{w}}_t, \tag{22}$$

where

$$\overline{b}(t, x) := (\nabla_x \cdot \Sigma)(T - t, x) + \Sigma(T - t, x)s(T - t, x) - b(T - t, x); \tag{23}$$

$$\overline{\sigma}(t, x) := \sigma(T - t, x) \tag{24}$$

$$\Sigma := \sigma\sigma^* \tag{25}$$

and $(\overline{\mathbf{w}}_t)_{t \in [0, T]}$ is another Wiener process. Since, by assumption, $\overline{\mathbf{x}}_T = \mathbf{x}_0$ is distributed according to our data distribution $\mu$, sampling from the data distribution can be achieved by simulating the SDE

(22). In practice, the usually unknown score $s$ is replaced by the score estimate $\tilde{s}$ learned during the training process.

### A.3 Integrating our forward process to the score-based framework

We can immediately use our forward process (11) for score-based generative modeling. To do so, we can interpret the forward process (11) as the solution of the SDE:

$$d\mathbf{y}_t = \beta_t\, dt + \varsigma_t\, d\mathbf{w}_t; \tag{26}$$

$$\mathbf{y}_0 = 0, \tag{27}$$

where

$$\beta_t := \frac{d}{dt} b_t \mathbf{x}_0; \tag{28}$$

$$\varsigma_t := \sqrt{2\tilde{\sigma}_t \frac{d}{dt}\sigma_t} \tag{29}$$

and

$$b_t := \sqrt{\bar{\alpha}_t}; \tag{30}$$

$$\sigma_t := \frac{\sqrt{1-\bar{\alpha}_t}}{(1-\tau(t))\sqrt{1+\frac{\|\nabla \mathbf{x}_0\|}{\lambda(t)}} + \tau(t)}; \tag{31}$$

$$\tilde{\sigma}_t := \sigma_t - \sigma_0. \tag{32}$$

However, it is more natural to translate our basic idea directly to an SDE and consider:

$$d\mathbf{y}_t = b_t\, dt + \sigma_t\, d\mathbf{w}_t; \tag{33}$$

$$\mathbf{y}_0 = \mathbf{x}_0 \tag{34}$$

instead. For the solution $(\mathbf{y}_t)_{t\in[0,\,T]}$ of an SDE of the form (33), $\mathbf{y}_t$ is conditionally Gaussian given $\mathbf{y}_0$. Assuming $\mathbf{y}_0$ is distributed according to the target data distribution $\mu$, we can use the general procedure described in Appendix A.1 and Appendix A.2 to train the score and sample from $\mu$.

## B Relation to Flow Matching

A recent advancement in the community of generative modeling is the framework of Flow Matching Lipman et al. (2022). Examining the framework of this paper, we believe that our edge-preserving noise scheduler fits naturally within their general framework for arbitrary functions $\mu(\mathbf{x}_1)$ and $\sigma_t(\mathbf{x}_1)$. Specifically, in their Equations (16)-(19), they derive a method for conditioning a vector field on the diffusion processes of Song and Ermon (2019) and Ho et al. (2020). Since our paper follows the same denoising probabilistic framework, and we chose $\gamma_t$ and $b$ consistent with Ho et al. (2020), the choice of $\mu(\mathbf{x}_1)$ of the reversed diffusion path can remain consistent with that of the variance-preserving path, while $\sigma_t(\mathbf{x}_1)$ would incorporate our noise-suppressing denominator inspired by Perona and Malik (1990). We believe the integration of our edge-preserving noise framework into Flow Matching is a promising direction for future work, motivated by their promising results on unconditional image generation, which we list in Table 2 below. Given our observations made in this paper, we are hopeful that an edge-preserving version of Flow Matching can further improve its performance, as we have seen with DDPM.

Table 2: Quantitative comparison for unconditional image generation between Flow Matching (Lipman et al., 2022) and Ours. We used the implementation available in the library torchcfm (with the 'ConditionalFlowMatcher' model). We provide the FID score of Ours (the variant of an edge-preserving process with our choices for $\gamma_t$ and $b$) as a relative comparison. To investigate the impact of edge-preserving noise, a more fair comparison would include an edge-preserving version of Flow Matching.

| Unconditional FID ($\downarrow$) | AFHQ-Cat($128^2$) | CelebA($128^2$) | LSUN-Church($128^2$) |
|---|---|---|---|
| Ours | 13.06 | 26.15 | 23.17 |
| Flow Matching | 7.43 | 14.5 | 12.86 |

## C  MOTIVATION BEHIND OUR HYBRID NOISE PROCESS

A valid observation to make is that given our hybrid forward process with two distinct stages, the edges are preserved longer, but still lost in the end. How does longer preservation of edges help the generative process? First thing to note is that the longer preservation of edges by itself does not have any impact, if we still let the model predict isotropic noise. Secondly, by modifying the forward process to be an edge-preserving one, the backward posterior formulation will also change and will rely on a non-isotropic variance, as discussed Eq. (13). *It is the combination of edge-preserving noise, together with our structure-aware loss function that makes the model work.* Furthermore, our frequency analysis (Section 5.1) has quantitatively shown that our decoupling approach is beneficial to learning the low-to-mid frequencies of the target dataset. This is consistent with recent work on wavelet-based diffusion models (Huang et al., 2024b), that demonstrates it is advantageous to learn low-frequency content separately from high-frequency content in the wavelet domain, using two distinct modules. Instead, we use two distinct stages, one that focuses on lower-frequency primary structural content (edge-preserving stage), and one that focuses on fine-grained high-frequency details (isotropic stage).

## D  HOW NEGATIVE LOG LIKELIHOOD CAN BE APPROXIMATED

Here we explain how negative log likelihood in the original DDPM Ho et al. (2020) paper can be approximated with our formulation.

The denoising probabilistic model paradigm defined in the DDPM paper defines the loss by minimizing a variational upper bound on the negative log likelihood. Because our noise is still Gaussian, the derivation they make in Eq. (3) to (5) of their paper still holds for us. The difference however is that we're non-isotropically scaling our noise based on the image content. As a result, our methods differ on Eq. (8) in their paper. Instead, we end up with the following form:

$$L_{t-1} = \mathbb{E}_q[\Sigma^{-1}(\tilde{\mu_t}(\mathbf{x}_t, \mathbf{x}0) - \mu\theta(\mathbf{x}_t, t)).(\tilde{\mu_t}(\mathbf{x}_t, \mathbf{x}0) - \mu\theta(\mathbf{x}_t, t))] \tag{35}$$

In essence, for our formulation that considers non-isotropic Gaussian noise, we need to apply a different loss scaling for each pixel.

Our formulation still provides an analytical variational upper bound to approximate the negative log-likelihood. While our heuristic loss function (Eq. (15)) already proved effective for approximating non-isotropic noise corresponding to structural content in the data, a more accurate KL-divergence loss would include the scaling discussed above.

## E  ADDITIONAL RESULTS

In this section, we provide additional results and ablations.

Table 3 shows quantitative FID comparisons using latent diffusion (Rombach et al., 2022) models on all the baselines.

Figure 9, Figure 10, Figure 11, Figure 12 and Figure 13 show more generated samples and comparisons between IHDM, DDPM on all previously introduced datasets. In Fig. 6 we show samples for the Human-Sketch ($128^2$) data set specifically. This dataset was of particular interest to us, given the images only consist of high-frequency, edge content. Although we observed that this data is remarkably challenging for all methods, our model is able to consistently deliver visually better results. Note that although we report FID scores for this data set, they are very inconsistent with the visual quality of the samples. This is likely due to the Inception-v3 backbone being designed for continuous image data, leading to highly unstable results when applied to high-frequency binary data.

Figure 7 shows an additional visualization of the impact $t_\Phi$ for the LSUN-Church ($128^2$) dataset. $t_\Phi = 0.5$ works best in terms of FID metric, consistent to the results shown in Section 5.2.

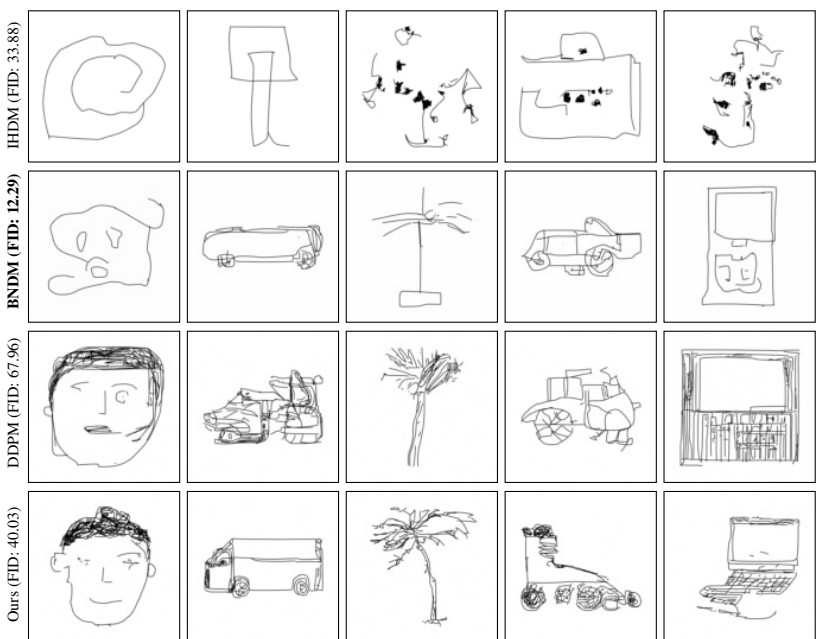

Figure 6: Generated unconditional samples for the Human Sketch ($128^2$) dataset (Eitz et al., 2012). All models were trained for an equal amount of 575 epochs. Note that the FID scores are inconsistent with visual quality. The cause for this is the Inception-v3 backbone, which is designed for continuous image data, leading to highly unstable results when applied to high-frequency binary data like hand-drawn sketches.

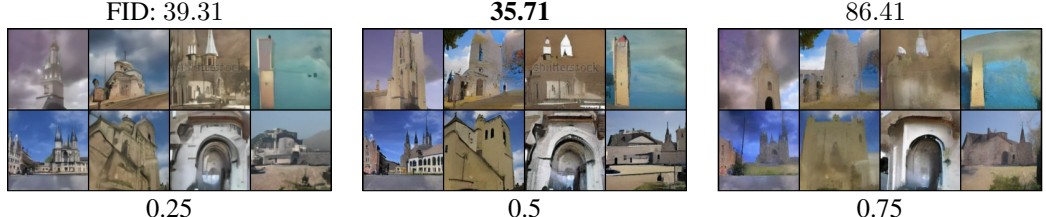

Figure 7: Impact of location of transition point $t_\Phi$ on sample quality, shown for the LSUN-Church ($128^2$) dataset. If we place $t_\Phi$ too far, the model happens to learn only the lowest frequencies and generates no details at all. Placing it too early leads to results that are less sharp. We found that by placing $t_\Phi$ at 50%, we strike a good balance between the two, leading to better quantitative and qualitative results.

Table 3: Quantitative FID score comparisons on latent diffusion models (Rombach et al., 2022) among IHDM (Rissanen et al., 2023), DDPM (Ho et al., 2020), BNDM (Huang et al., 2024a) and our method.

| Unconditional FID ($\downarrow$) | CelebA($256^2$, latent) | AFHQ-Cat($512^2$, latent) |
|---|---|---|
| IHDM | 88.12 | 28.09 |
| DDPM | **7.87** | 22.86 |
| BNDM | 10.93 | **13.62** |
| Ours | 13.89 | 18.91 |

Table 4: We performed unconditional image generation experiments on an unofficial implementation of SimpleDiffusion (Hoogeboom et al., 2023). Note that this implementation is missing the architectural changes proposed in their paper. We are using the same simple architecture as Ours. As a result, the FID numbers are in double-digit range and comparable to ours. We contend that to measure the impact of edge-preserving noise, it would be more fair to compare against an edge-preserving version of Simple Diffusion using the framework presented in this work.

| Unconditional FID ($\downarrow$) | CelebA($128^2$) | LSUN-Church($128^2$) | AFHQ-Cat($128^2$) |
|---|---|---|---|
| Ours | 26.15 | 23.17 | **13.06** |
| Simple Diffusion | **19.28** | **17.87** | 15.66 |

Table 5: Additional results for the unofficial implementation of SimpleDiffusion (Hoogeboom et al., 2023) applied to the shape-guided generative task according to SDEdit (Meng et al., 2022). Note that this unofficial version does not include the architectural changes proposed in the paper. While the version of our method presented in this paper already outperforms Simple Diffusion in the majority of test cases, we again would like to make the comment that a comparison with an edge-preserving version of Simple Diffusion would be more fair. We consider the integration of our edge-preserving noise framework into other works, like Simple Diffusion, a promising direction for future work.

| Shape-guided task FID ($\downarrow$) | CelebA($128^2$) | LSUN-Church($128^2$) | AFHQ-Cat($128^2$) |
|---|---|---|---|
| Ours | **39.08** | 56.14 | **23.50** |
| Simple Diffusion | 46.43 | **45.93** | 34.26 |

Table 6: Our measurements on time and memory consumptions are based on data resolution (128x128) and a batch size of 64. Note that BNDM and Flow Matching make use of less inference steps (T=250 vs. T=500 for Ours, DDPM and Simple Diffusion), and therefore are expected to be faster for inference. Given that the official implementations of Simple Diffusion (Hoogeboom et al., 2023) and Flow Matching (Lipman et al., 2022) are unavailable, we used simpleDiffusion and torchcfm respectively. Our setup consisted of 2 NVIDIA Quadro RTX 8000 GPUS. We see that timings and memory usage of Ours is very similar to DDPM and Simple Diffusion, suggesting that the Sobel filter we apply to approximate brings minimal overhead.

| | Ours | DDPM | BNDM | Flow Matching | Simple Diffusion |
|---|---|---|---|---|---|
| Training time (seconds per iteration) | 1.12 | 1.11 | 0.74 | 2.74 | 1.48 |
| Inference time (seconds per iteration) | 301.5 | 277.5 | 77.2 | 84.78 | 290.7 |
| Inference Memory (GB) | 9.16 | 9.16 | 10.3 | 22.18 | 10.42 |

Table 7: Shape-guided image generation (based on SDEdit (Meng et al., 2022)): precision (metric for realism) and recall (metric for diversity) scores (Kynkäänniemi et al., 2019) for isotropic model DDPM, and our edge-preserving model. We consistently outperform in terms of precision, and again closely match in terms of recall.

| | Ours | | DDPM | |
|---|---|---|---|---|
| Shape-guided image generation | Precision ($\uparrow$) | Recall ($\uparrow$) | Precision ($\uparrow$) | Recall ($\uparrow$) |
| AFHQ-Cat($128^2$) | **0.93** | **0.80** | 0.92 | 0.66 |
| CelebA($128^2$) | **0.65** | 0.46 | 0.53 | **0.53** |
| LSUN-Church($128^2$) | **0.87** | 0.46 | 0.84 | **0.50** |

Table 8: Unconditional image generation: precision (metric for realism) and recall (metric for diversity) scores for isotropic model DDPM, and our edge-preserving model. While we slightly get outperformed, we find that our edge-preserving model closely matches DDPM on both metrics. therefore we would argue that edge-preserving noise minimally impacts diversity.

| Unconditional image generation | Ours | | DDPM | |
|---|---|---|---|---|
| | Precision ($\uparrow$) | Recall ($\uparrow$) | Precision ($\uparrow$) | Recall ($\uparrow$) |
| AFHQ-Cat($128^2$) | 0.76 | 0.20 | **0.77** | **0.21** |
| CelebA($128^2$) | 0.90 | 0.16 | **0.92** | **0.17** |
| LSUN-Church($128^2$) | **0.65** | 0.33 | 0.47 | **0.38** |

Table 9: We additionally integrate our method into RePaint (Lugmayr et al., 2022), a state-of-the-art pixel-space inpainting algorithm. We made a comparative analysis between RePaint and "Edge-preserving RePaint" by performing an inpainting task over 100 images for multiple datasets. Visual results for this task are shown in Fig. 14. Note that lower FID is better, and higher CLIP Score (Radford et al., 2021) is better. We find that our model closely matches the performance of RePaint.

| FID/CLIP | Ours | RePaint |
|---|---|---|
| AFHQ-Cat($128^2$) | 20.50/97.91 | 19.77/98.21 |
| CelebA($128^2$) | 49.12/91.58 | 44.95/93.64 |

Table 10: We provide additional comparison for our shape-guided generative task (Meng et al., 2022) evaluated using the CLIP metric (Radford et al., 2021). Our method consistently outperforms the baselines on this metric, indicating that the generated images are more semantically aligned with the ground-truths (the original images used to generate the stroke paintings). We show several examples (Fig. 5 and Fig. 8) where our model solves visual artifacts that are apparent with other baselines, which can improve the semantical meaning of the generated image.

| CLIP | Ours | DDPM | Simple Diffusion |
|---|---|---|---|
| AFHQ-Cat($128^2$) | **88.97** | 88.78 | 88.23 |
| CelebA($128^2$) | **61.15** | 61.02 | 60.72 |
| LSUN-Church($128^2$) | **64.32** | 62.57 | 62.13 |

Figure 8: More samples for our model and other baselines applied to SDEdit (Meng et al., 2022). Note how our model is able to generate sharper results that suffer less from artifacts. Although BNDM can generate satisfactory results in certain cases (e.g., cat and church), it often deviates from the stroke painting guide, potentially producing outcomes that differ significantly from the user's original intent. In contrast, our method closely follows the stroke painting guide, accurately preserving both shape and color.

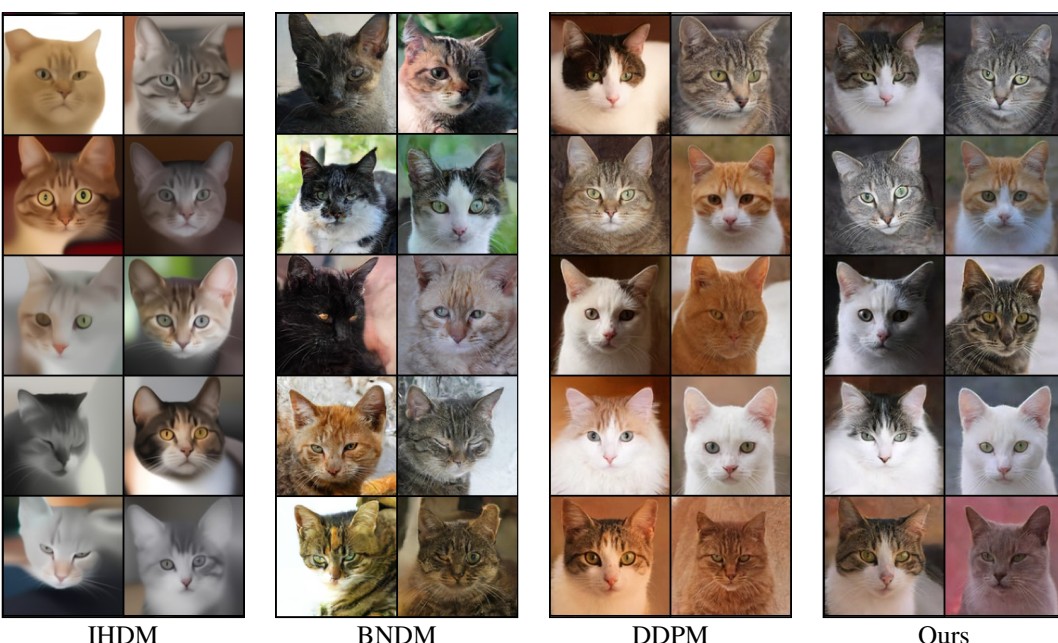

IHDM         BNDM         DDPM         Ours

Figure 9: More unconditional samples for IHDM, DDPM and our method on the AFHQ-Cat ($128^2$) dataset. Although the difference between DDPM and our method is subtle, we consistently found that our approach captures geometric details more effectively (e.g., whiskers) and experiences fewer blurry artifacts (e.g., right sample in row 3, DDPM vs. Ours).

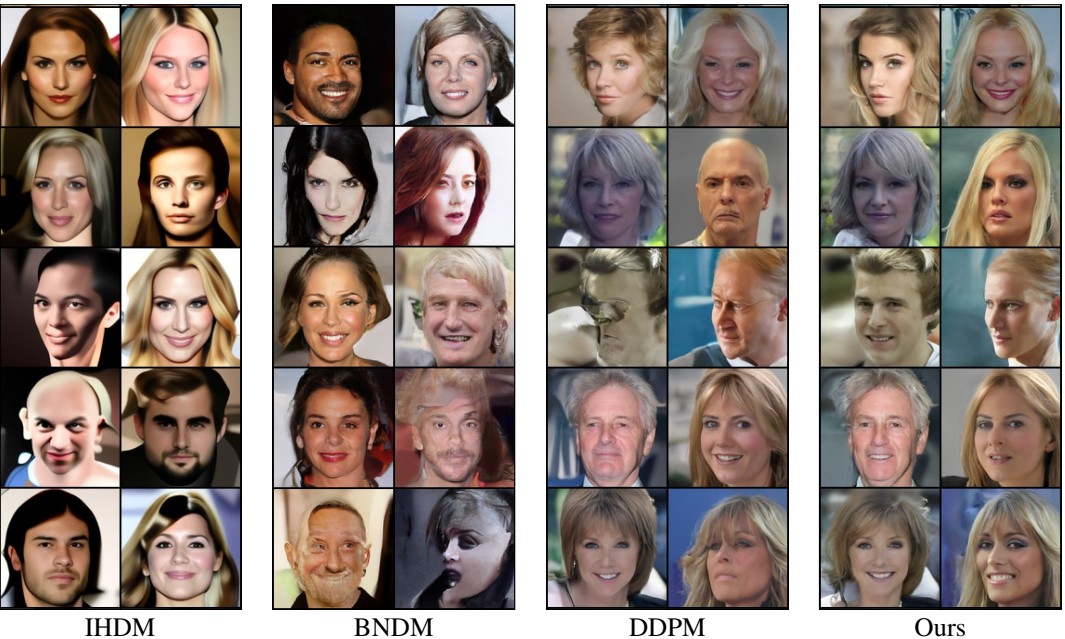

IHDM         BNDM         DDPM         Ours

Figure 10: More unconditional samples for IHDM, BNDM, DDPM and our method on the CelebA ($128^2$) dataset. While BNDM is only slightly outperformed by our model in terms of FID metric, its samples look noticeably different in terms of colors. We attribute this difference to the fact that BNDM simulates an ODE, where we in contrast simulate an SDE, possibly causing both methods to sample a different part of the manifold. In terms of visual quality the BNDM samples also show more artifacts, but it is known from previous work that FID score does not always well reflect percepted quality (Liu et al., 2018).

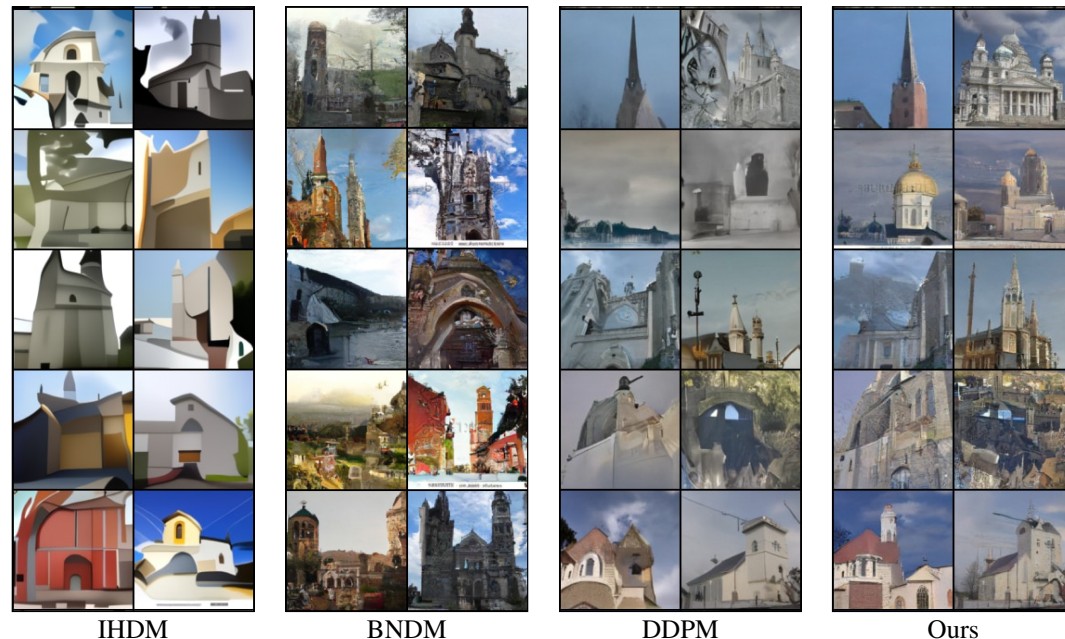

|       |       |      |      |
|-------|-------|------|------|
| IHDM  | BNDM  | DDPM | Ours |

Figure 11: More unconditional samples for IHDM, BNDM, DDPM and our method on the LSUN-Church ($128^2$) dataset. lthough our results appear similar to DDPM's, our method more effectively captures the geometric details of buildings and exhibits fewer artifacts, such as blurry regions, compared to DDPM.

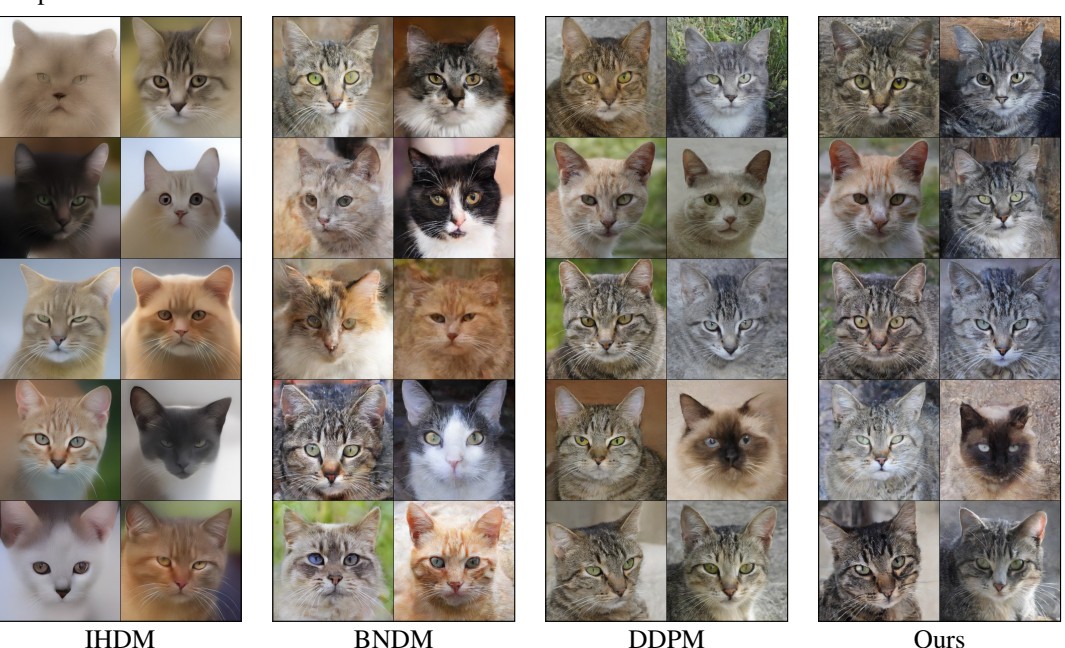

|       |       |      |      |
|-------|-------|------|------|
| IHDM  | BNDM  | DDPM | Ours |

Figure 12: More unconditional samples for IHDM, DDPM and our method on the AFHQ-Cat ($512^2$, LDM) dataset. All samples are generated via diffusion in latent space. Note that despite the deficit in FID score, our method is able to produce results of very similar perceptual quality.

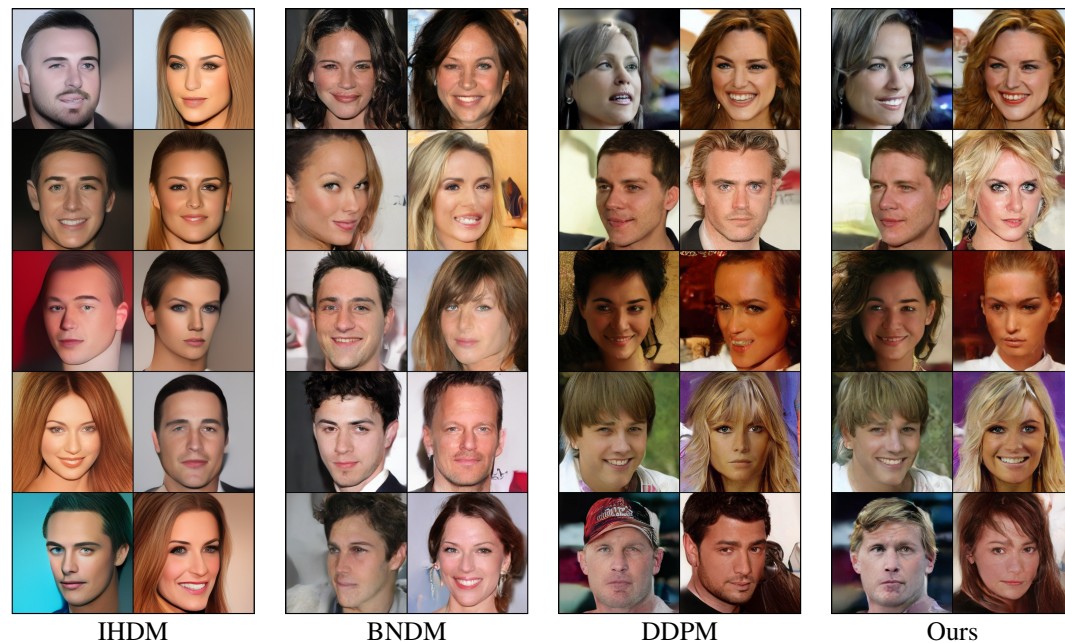

| IHDM | BNDM | DDPM | Ours |

Figure 13: More unconditional samples for IHDM, DDPM and our method on the CelebA ($256^2$, LDM) dataset. All samples are generated via diffusion in latent space. Although our method is slightly outperformed in terms of the FID metric, the visual quality of our samples is highly comparable to the baselines, and in some cases, even superior (e.g., third row of DDPM vs. Ours).

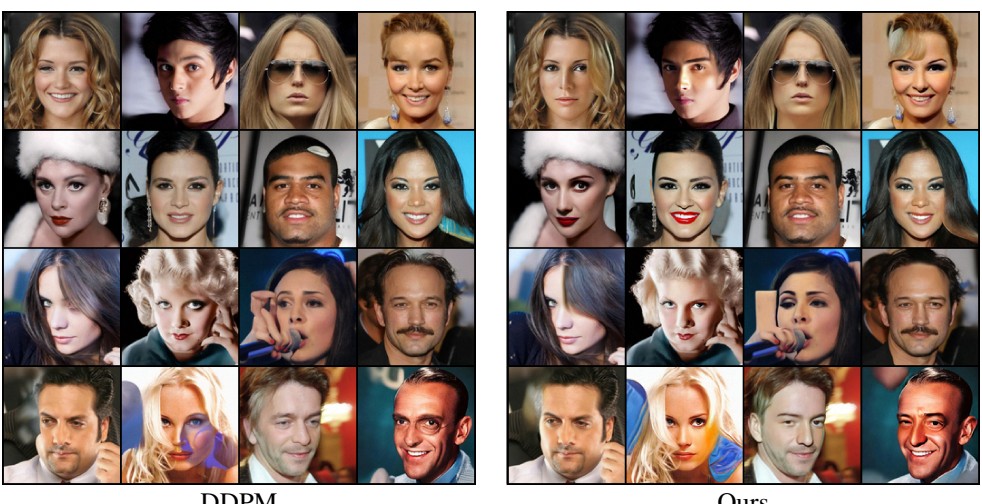

| DDPM | Ours |

Figure 14: Visual comparisons on RePaint (Lugmayr et al., 2022) using the isotropic DDPM model and our edge-preserving noise model. We find that our model closely matches the performance of RePaint.

