# OpenReview forum: "Edge-preserving noise for diffusion models"
_ICLR.cc/2025/Conference — Submitted to ICLR 2025_

### Official Review · Reviewer_Q9qP · 2024-11-01

**Soundness:** 2
**Presentation:** 2
**Contribution:** 3
**Rating:** 6
**Confidence:** 3

**Summary:**

The authors proposed a variant of DDPM, named content-aware diffusion models, putting more focus on the edges. They made some modifications on the noise scheme to make it edge-preserving noise, and edit the formula for forward and backward calculation as well.

**Strengths:**

The authors proposed an interesting idea about how to keep the content clear and stable during the diffusion process, assuming the loss of detail content is caused by the isotropic noise scheme (like DDPM), by adopting existing research ideas from image processing area.

**Weaknesses:**

1. Experimentations might not be enough, especially in the quantitative comparisons with other existing works. Especially, if you want to state your proposed methods can be more content-aware compared to “isotropic diffusion models”, but not ONLY “DDPM”, then you have to compare it with other recent “isotropic diffusion models”.
2. In addition, although with good explanation and math works, I’m feeling like this adaptation is more like adding an “weight matrix” to a specific focus, via an existing formula learned from image processing domain, during the diffusion process, and thus might not be impactful/novel enough. I would like to also hear other reviewers’ comments.

**Questions:**

1. “While the visual improvement over DDPM is subtle, our model generally demonstrated greater robustness to artifacts.” May I ask where can I see the experimentation results support this statement?
2. “In practice, we didn’t notice any increase in training/inference time.” May I ask what are the approximate numbers for the training/inference time?

---

> ### Author Response · Authors · 2024-11-22
> **Response to reviewer Q9qP (Part 1)**
>
> We provide additional comparisons with the SOTA Simple Diffusion (Hoogeboom et al., 2023), being outperformed in unconditional tasks but consistently outperforming in shape-guided generation.
>
> For shape-guided image generation, we also consistently outperform this state-of-the-art pixel-space isotropic baseline:
>
> | FID/CLIP score - Shape-guided image generation (SDEdit (Meng et al. 2021))| Ours                  | Simple Diffusion  |
> |---------------------------------------------------------------------------|-----------------------|-------------------|
> | AFHQ-Cat($128^2$)                                                         | **23.50** / **88.97** | 34.26 / 88.23     |
> | CelebA($128^2$)                                                           | **39.08** / **61.15** | 46.43  / 60.72    |
> | LSUN-Church($128^2$)                                                      | 56.14 / **64.32**     | **45.93** / 62.13 |
>
>  Although we get outperformed in unconditional image generation, we are optimistic that incorporating edge-preserving noise into Simple Diffusion will enhance its performance, as we saw with DDPM. We contend that these are reasonable expectations, as both our method and theirs align with the denoising probabilistic paradigm of known Gaussian processes outlined in paragraph 1 of the global response sent to all reviewers.
>
> | FID - Unconditional image generation | Ours  | Simple Diffusion (Hoogeboom et al. 2023) |
> |--------------------------------------|-------|------------------------------------------|
> | AFHQ-Cat($128^2)                     | 13.06 | 15.66                                    |
> | CelebA($128^2$)                      | 26.15 | 19.28                                    |
> | LSUN-Church($128^2)                  | 23.17 | 17.87                                    |

---

> ### Author Response · Authors · 2024-11-22
> **Response to reviewer Q9qP (Part 2)**
>
> ## 2. This adaptation seems more like adding a "weight matrix" to a specific focus, using existing work from the image processing domain.
> The goal of our work is to investigate the impact of edge-preserving noise on a generative diffusion process, which is inspired by the idea of edge-preserving denoising introduced by Perona and Malik in image processing. Despite this, we had to make several efforts to make this work in the context of generative diffusion:
>
> 1. Their work considers an anisotropic diffusion process described in Eq. (10) of our submission. Unfortunately, Eq. (10) cannot be written as a linear equation of the form Eq.(1) because of its dependency on $||\nabla \mathbf{x}_t||$. By considering an alternative form of the diffusion coefficient in Eq. (11) such that  it incorporates $||\nabla \mathbf{x}_0||$ instead, we are now able to use it in a linear diffusion process.
>
> 2. With our updated forward process, the forward and backward posteriors of the generative process change as well, as described in Eq. (4) to (8) in the paper.
>
> 3. Our backward process posteriors depend on non-isotropic variance (see Eq. (14) - (16) in our submission), introduced by the edge-preserving schedule. Therefore, we had to rethink the loss function in order to make sure the model learns this non-isotropic variance corresponding to the structural edge content in the data.
>
> 4. We cannot simply apply linearly varying edge-preserving noise, but needed to design a hybrid process that interpolates between edge-preserving and isotropic noise, to ensure that our process converges to the known prior $\mathcal{N}(0, I)$ from which we can start the sampling.
>
> 5. Another crucial component is to ensure smoothness in the transition between edge-preserving and isotropic noise. In the ablation in Section 5.2 of our submission we show that choosing \tau(t) to be linear instead of a steeper function like sigmoid or cosine has a great impact on performance. Additionally, by using constant values for edge sensitivity $\lambda(t)$ instead of a smoother interpolation, the model focuses on a specific granularity of structural content but fails to generate finer details.
>
> Therefore, we would like to contend that there is more going on than simply applying a weight matrix. Our edge-preserving process fits into the two main general paradigms of generative diffusion models.
>
> In Practice, our work shows its **positive impact** as follows:
>
> - In particular, our method significantly outperforms the baselines on shape-guided generative tasks (stroke painting to image), both quantitatively but especially qualitatively (sharper reconstructions, less artifacts).
> - We demonstrate that our edge-preserving model is better able to learn low-to-mid frequencies of the target data distribution.
> - We show quantitative and qualitative improvements w.r.t. state-of-the-art isotropic models (DDPM, Simple Diffusion) for **unconditional image generation**.
> - We show how it fits in 2 general mathematical frameworks: that of a well-studied Gaussian process (Eq. (4) to (8) in the paper), and that of an SDE , allowing it to be adopted in various diffusion settings. Given these 2 general interpretations we also believe that our process should be integrable in the Flow Matching (Lipman et al. 2023) framework.
> - We will release minimalistic and modular source code that is easy to integrate or extend upon publication.
>
> ## 3. Which results support the greater robustness to visual artifacts compared to DDPM?
> The visual artifacts that we mention in this statement occur ocassionally, and given that we showed uncurated samples in the submission, they might be very subtle or missing. For the shape-guided task, we have seen them occur more often and being more pronounced (as is shown in Figures 5 and 8). However, for unconditional generation, one extreme example can be found in Figure 10, row 3 in the first column for ours vs. DDPM. We plan to provide more specific examples in the revised version which we will upload later during this rebuttal phase to support this statement.
>
> ## 4. What are the approximate numbers for training and inference time?
> Please find our response to this concern in paragraph 4 of the global response we sent to all reviewers. We will incorporate these discussions in the revised version.

---

> > ### Comment · Reviewer_Q9qP · 2024-11-26
> >
> > Thank the authors for the detailed reply. I have updated my score accordingly.

---

### Official Review · Reviewer_WK9j · 2024-11-01

**Soundness:** 2
**Presentation:** 3
**Contribution:** 2
**Rating:** 5
**Confidence:** 3

**Summary:**

The paper modifies the original DDPM formulation by Ho et al., altering the noise schedule to concentrate in localized regions according to the gradient magnitude of the image $\frac{|| \nabla_{\mathbf{x}_0}||}{\lambda(t)}$ and edge sensitivity $\lambda(t)$, offering control over different frequencies modelled during training. This approach leads to up to improvement in FID over the DDPM baseline, and the results qualitatively demonstrate finer details and better structure can be modeled.

**Strengths:**

The idea is intuitively simple and straightforward to incorporate into generative imaging applications. It has the potential to offer fine-grained control over the geometric structure captured in the diffusion process. Quantitative and qualitative improvements are demonstrated over baselines across several datasets.

**Weaknesses:**

Substantial advances in the diffusion community have not been discussed or incorporated adequately within this paper. Therefore, it is not clear to the community whether this approach will offer significant benefit when scaling to larger problems or stronger baselines. In particular:

* The reported FIDs and qualitative results do not seem competitive with current state-of-the-art diffusion models. FIDs of around **26.15** for 128$\times$128 seem too high. Zhang et al., 2022, "Dimensionality-Varying Diffusion Process", attain CelebA 128^2 of around **~6**, $\infty-$Diff (Bond-Taylor et al., ICLR 2024) receives FID_CLIP on CelebAHQ of ~3, and recent diffusion papers attain ~2 on even more difficult datasets such as ImageNet (see the review in Hoogeboom et al., "Simpler Diffusion" Table 3, covering low FID 2023/2024 papers).
* Recent advances in Flow Matching (Lipman et al., ICLR 2023) offer a simpler theoretical framework, offering a direct (near optimal) path through the transition space—demonstrated in real-world settings such as StableDiffusion3. This could simplify the edge-preserving schedule, and make it easier to control than Ho el al., schedule derivation, while giving improved samples. Likelihood estimates can also be attained through FM. I would like to see discussion or comparison.

**Questions:**

* It is not clear whether NLLs can be approximated easily with this approach, potentially limiting the applications. Please clarify or demonstrate this.
* The time-varying edge sensitivity $\lambda(t)$ seems arbitrary in section 5, and is currently dependent on Ho et al.,'s schedule, tying this down to similar variants.
   * Can the main theory be better abstracted and demonstrated in different diffusion frameworks?
   * Is the approach suited for Flow Matching?
* Implementing this within a modern diffusion framework/architecture that efficiently supports high-resolution datasets would be beneficial to quantitatively evaluate improvements.
   * How does this scale with FFHQ $1024^2$ or ImageNet-$128$?
* Please clarify how $|| \nabla_{\mathbf{x}_0}||$ is implemented. Is this approximated with a Sobel filter?

---

> ### Author Response · Authors · 2024-11-22
> **Response to reviewer WK9j (Part 1)**
>
> Dear reviewer, thank you for your valuable feedback and suggestions. Besides this response directed to you, we have also prepared a global response to all reviewers, discussing common concerns/questions and providing further details.
>
> ## 1. The reported FIDs and qualitative results do not seem competitive with current state-of-the-art diffusion models.
> In paragraph 3 of the global response sent to all reviewers, we provide an additional comparison with Simple Diffusion (Hoogeboom et al. 2023) [6]. We also provide CLIPScore as a new metric for these comparisons. We see that our model still consistently outperforms this baseline on the shape-guided generative task. We also discuss there why our reported FIDs seem to be on the higher side. We would like to refer you to paragraph 3 of the global response for the actual scores, as well as additional details.
>
> ## 2. How does the method compare to and fit in the framework of Flow Matching?
> Thank you for the suggestion. We find this an exciting direction for future work. After a closer examination of the Flow Matching paper, we believe our edge-preserving noise scheduler fits naturally within their general framework for arbitrary functions $\mu_t(x_1)$ and $\sigma_t(x_1)$. Specifically, in their Equations (16)–(19), they derive a method for conditioning a vector field using Gaussian processes defined in [4] and [8]. Since our method operates within the same Gaussian process framework, the choice of $\mu_t(x_1)$ or the reversed diffusion path can remain consistent with that of the variance-preserving path, while $\sigma_t(x_1)$ would incorporate our noise-suppressing denominator inspired by Perona and Malik.
>
> Based on your request, we also made a quantitative comparison and qualitative with Flow Matching. Given that an official implementation by the authors is unavailable, we used the popular library [torchcfm](https://github.com/atong01/conditional-flow-matching), which contains an implementation for [4]. We performed an equal time training w.r.t. our edge-preserving model.
>
> | FID - Unconditional image generation | Ours  | Flow Matching (Lipman et al. 2023) |
> |--------------------------------------|-------|------------------------------------|
> | AFHQ-Cat($128^2)                     | 13.06 | 7.43                               |
> | CelebA($128^2$)                      | 26.15 | 14.5                               |
> | LSUN-Church($128^2)                  | 23.17 | 12.86                              |
>
> **Given Flow Matching’s proven effectiveness in the results above, combining it with our edge-preserving noise scheme holds promise for further improving sample quality.**
>
> ## 3. Can the negative log likelihood still be approximated (via variational upper bound) with this formulation?
> Thank you for pointing this out, this is an interesting question. The denoising probabilistic model paradigm defined in the DDPM paper defines the loss by minimizing a variational upper bound on the negative log likelihood. Because our noise is still Gaussian, the derivation they make in Eq. (3) to (5) of their paper still holds for us. **The difference however is that we're non-isotropically scaling our noise based on the image content. As a result, our methods differ on Eq. (8) in their paper.** Instead, we will end up with something of the form $L_{t-1} = \mathbb{E}_q[\Sigma^{-1}(\tilde{\mathbf{\mu_t}}(\mathbf{x}_t, \mathbf{x}_0) - \mathbf{\mu_{\theta}}(\mathbf{x}_t, t)) . (\tilde{\mathbf{\mu_t}}(\mathbf{x}_t, \mathbf{x}_0) - \mathbf{\mu_{\theta}}(\mathbf{x}_t, t))]$ In essence, for our formulation that considers non-isotropic Gaussian noise, one should apply a different loss scaling for each pixel.
>
> So **yes, our formulation still provides an analytical variational upper bound to approximate the negative log-likelihood.** While our heuristic loss function (Eq. 17) already proved effective for approximating non-isotropic noise corresponding to structural content in the data, a more accurate KL-divergence loss would include the scaling discussed above. We will add a more detailed discussion on this in the Appendix of the updated version.
>
> ## 4. The time-varying edge sensitivity $\lambda(t)$ seems arbitrary and is currently dependent on Ho et al.'s schedule.
> We will add a brief note to the ablation (Section 5.2) mentioning that values $\lambda_{min}$ and $\lambda{max}$ described in Section 5 were chosen based on experimentation in the range $1e^{-1}$ and $1e^{-8}$, but had less impact on performance than other parameters. We would like to clarify however that $\lambda(t)$ does not depend on DDPM's schedule. In fact, we would like to emphasize that our method is completely independent of DDPM, as is discussed in paragraph 2 of the global response to all reviewers.

---

> ### Author Response · Authors · 2024-11-22
> **Response to reviewer WK9j (Part 2)**
>
> ## 5. Please clarify how $||\nabla x_0||$ is implemented.
> Currently, our implementation indeed considers the approximation of $||\nabla x_0||$ made with a Sobel filter. As an alternative, we have also tried to use a Canny edge detector as a signal to determine where to suppress the noise, but this proved significantly less effective.
>
> ## 6. Implementing this within a diffusion framework that efficiently supports high-resolution datasets would be beneficial for quanititative evaluation
> Integrating our edge-preserving model into high-resolution synthesis frameworks like Simple Diffusion (Hoogeboom et al. 2023) [6] is a promising and realistic direction for future work, given that both models are based on the same Gaussian process mathematical framework outlined in Section 3 of our paper.
>
> Something that does need to be taken care of when moving to higher resolutions however is the computation of $||\nabla x_0||$. Our current implementation uses a Sobel filter, which is not scale-invariant. For the lower resolution data (up to 128x128) we have tested on we have found that this does not pose a real problem. For higher resolution data, this can become an issue however, and it might be better to resort to a scale-invariant "edge detector" like a Laplacian of Gaussian (LoG) with a scale-normalization strategy.
>
> ## 7. Can the theory be better abstracted and therefore applied inside other diffusion frameworks?
> Thank you for this interesting question. In paragraph 1 of the global response to all reviewers, we show how our method fits in the two main general paradigms of generative diffusion models. In particular, we mention that we follow the probabilistic paradigm in our submission. Besides this our Eq. (13) can be reformulated as an SDE, following the score-based paradigm. We would like to refer to paragraph 1 of the global response for more concrete details on this.

---

> > ### Comment · Reviewer_WK9j · 2024-11-26
> >
> > Thank you for your reply. I remain uncertain, particularly given the weak FIDs, and vanilla flow matching outperforms your inductive bias, in terms of both FID and sampling speed. The timing/memory comparisons could be more rigorously presented. For example, a Sobel filter has negligible impact here—memory usage is primarily determined by the attention/U-Net architecture, which varies across the methods compared and is not inherently tied to your inductive bias. I think a more meaningful evaluation would isolate the effect on your inductive bias by integrating it into existing frameworks (where possible) and directly comparing the memory, quality and timing within the same architecture and diff.eq solver parameterization. This would provide clear insight for the community on whether incorporating your inductive bias is worthwhile for future diffusion implementations as the field evolves.

---

> > > ### Author Response · Authors · 2024-11-27
> > > **Response to reviewer WK9j**
> > >
> > > Thank you for your response. We fully agree that the impact of our inductive bias would be more effectively demonstrated when our noise model is integrated into a specific framework.
> > >
> > > Currently, the comparison shown in the table with flow matching is not entirely fair and can be likened to comparing apples to oranges. A more appropriate comparison would involve incorporating our edge-preserving noise into the flow matching model. However, due to the limited time available during the rebuttal phase, we were unable to complete this implementation. But we have outlined:
> > > - in Appendix B of our revised paper, how our framework can be integrated into the flow matching framework.
> > > - the same considerations apply to SDE-based or score-based generative models.
> > >
> > > At present, our model is implemented only within the denoising probabilistic model (DDPM) framework. Since vanilla flow matching outperforms vanilla DDPM, it is unsurprising that vanilla flow matching also outperforms our approach in its current form. We kindly ask reviewers to consider this context before interpreting the comparison between Ours vs vanilla Flow matching.
> > >
> > > Overall, we are excited about the potential for our edge-preserving noise model to be integrated into various diffusion model frameworks in the near future. Our updated revision includes some insights into how our model can be incorporated into different frameworks (see Appendices A and B).

---

### Official Review · Reviewer_mUvY · 2024-11-02

**Soundness:** 3
**Presentation:** 3
**Contribution:** 3
**Rating:** 6
**Confidence:** 3

**Summary:**

This paper introduces a novel edge-preserving diffusion model that generalizes denoising diffusion probabilistic models (DDPMs). By incorporating an edge-aware noise scheduler, the model retains structural information in images, converges more rapidly to the target distribution, and better captures low-to-mid frequency components within datasets, which are crucial for representing shapes and structural details.

**Strengths:**

This paper proposes an interesting approach to improve the generation quality of diffusion models.
The experiments are extensive, and the visualizations are clear, helping readers understand the improved generation performance.

**Weaknesses:**

1. Does edge-aware noise affect generation diversity?
2. In the "Related Work" section, you initially describe IHDM as an isotropic Gaussian blurring model, yet later refer to it and BNDM as incorporating a form of non-isotropic noise. This creates a seeming contradiction that would benefit from further clarification to ensure the logical coherence of your manuscript.
3. While the paper makes a commendable effort to integrate the anisotropic diffusion concept from Perona and Malik (1990) into the diffusion model framework, the innovation appears to be primarily an application of existing theory rather than a novel theoretical contribution. The incorporation of edge-preserving noise as a new term in the diffusion process, while potentially enhancing the model's performance in certain aspects, does not fundamentally alter the underlying principles of generative diffusion models.
4. The discussion surrounding Eq. (2) and Eq. (3) in the paper is somewhat perplexing due to its circuitous approach. The authors explore the relationship between diffusion models and heat conduction, as well as the limitations of isotropic noise, before returning to the contributions of Perona and Malik (1990) in Eq. (10). A more effective strategy would be to engage with Perona and Malik's work on edge-preserving diffusion earlier in the discussion. This would create a natural transition to Eq. (12), which introduces the paper's core innovation—a novel noise term.
5. In the "Background" section, when discussing the "Forward and backward process posteriors," you utilize the formulation presented in Eq. (1), which is consistent with the isotropic Gaussian process. However, you also reference Eq. (2), which is derived from the heat dissipation process and represents a different formulation. My concern is that the use of both Eq. (1) and Eq. (2) in this context may create confusion, especially since your novel edge-preserving noise scheme seems to be more aligned with the principles of Eq. (1), as inferred from the derivations in Section 4.2. It appears that your improved noise term is inspired by the anisotropic diffusion work of Perona and Malik, which is more closely related to Eq. (2). Yet, the parameter calculations and posterior calculations in your model follow a form similar to Eq. (1). If this is the case, then the inclusion of Eq. (2) in the introduction might be seen as unnecessary, and it could potentially detract from the clarity of your paper. I would recommend either providing a more explicit connection between your noise model and Eq. (2), or if Eq. (2) is not central to your approach, consider streamlining the presentation to focus on Eq. (1) and its relevance to your edge-preserving noise scheme. This would help readers to better understand the relationship between your work and the existing literature and clarify why both equations are presented if they are not both integral to your model.

**Questions:**

Please refer to the Weakness section.

---

> ### Author Response · Authors · 2024-11-22
> **Response to reviewer mUvY (Part 1)**
>
> Dear reviewer, thank you for your valuable feedback and suggestions. Besides this response directed to you, we have also prepared a global response to all reviewers, discussing common concerns/questions and providing further details.
>
> ## 1. Does edge-aware noise affect generation diversity?
> To answer your question and measure the mere impact of edge-preserving noise on diversity, we have computed precision (metric for realism) and recall (metric for diversity) scores for the isotropic model DDPM, and our edge-preserving model. We provide tables with these metrics below. For the shape-guided generative tasks (based on SDEdit (Meng et al. 2021)), we consistently outperform in terms of precision, and again closely match  in terms of recall.
>
> | Precision score - Shape-guided image generation (SDEdit) | Ours     | DDPM |
> |----------------------------------------------------------|----------|------|
> | AFHQ-Cat($128^2$)                                        | **0.93** | 0.92 |
> | CelebA($128^2$)                                          | **0.65** | 0.53 |
> | LSUN-Church($128^2$)                                     | **0.87** | 0.84 |
>
> | Recall score - Shape-guided image generation (SDEdit) | Ours     | DDPM     |
> |-------------------------------------------------------|----------|----------|
> | AFHQ-Cat($128^2$)                                     | **0.80** | 0.66     |
> | CelebA($128^2$)                                       |   0.46   | **0.53** |
> | LSUN-Church($128^2$)                                  |   0.46   | **0.50** |
>
> For unconditional image generation, while we slightly get outperformed, we find that our edge-preserving model closely matches DDPM on both metrics. therefore we would argue that edge-preserving noise minimally impacts diversity.
>
> | Precision score - Unconditional image generation | Ours     | DDPM     |
> |--------------------------------------------------|----------|----------|
> | AFHQ-Cat($128^2$)                                | 0.76     | 0.77 |
> | CelebA($128^2$)                                  |   0.90   |  0.92 |
> | LSUN-Church($128^2$)                             | **0.65** |   0.47   |
>
> | Recall score - Unconditional image generation | Ours | DDPM     |
> |-----------------------------------------------|------|----------|
> | AFHQ-Cat($128^2$)                             | 0.20 | 0.21 |
> | CelebA($128^2$)                               | 0.16 | 0.17 |
> | LSUN-Church($128^2$)                          | 0.33 | 0.38 |
>
> ## 2. Seeming contradiction about IHDM being a(n) (non-)isotropic Gaussian model
> Thanks for pointing this out. Although IHDM is a model that blurs isotropically in the **image space**, the diffusion is done in **frequency space** by taking a discrete cosine transform, and **it is in this frequency space that they consider a non-isotropic form of noise**. In short, the isotropic blurring in image space is equivalent to a non-isotropic addition of noise in the frequency space. We will be more explicit about this in the revised version of our paper that we will upload later during this rebuttal phase.

---

> ### Author Response · Authors · 2024-11-22
> **Response to reviewer mUvY (Part 2)**
>
> ## 3. The innovation appears to be primarily an application of existing theory rather than a novel theoretical contribution.
> The goal of our work is to investigate the impact of edge-preserving noise on a generative diffusion process, which is inspired by the idea of edge-preserving denoising introduced by Perona and Malik in image processing. Despite this, we had to make several efforts to make this work in the context of generative diffusion:
>
> 1. Their work considers an anisotropic diffusion process described in Eq. (10) of our submission. Unfortunately, Eq. (10) cannot be written as a linear equation of the form Eq.(1) because of its dependency on $||\nabla \mathbf{x}_t||$. By considering an alternative form of the diffusion coefficient in Eq. (11) such that  it incorporates $||\nabla \mathbf{x}_0||$ instead, we are now able to use it in a linear diffusion process.
>
> 2. With our updated forward process, the forward and backward posteriors of the generative process change as well, as described in Eq. (4) to (8) in the paper.
>
> 3. Our backward process posteriors depend on non-isotropic variance (see Eq. (14) - (16) in our submission), introduced by the edge-preserving schedule. Therefore, we had to rethink the loss function in order to make sure the model learns this non-isotropic variance corresponding to the structural edge content in the data.
>
> 4. We cannot simply apply linearly varying edge-preserving noise, but needed to design a hybrid process that interpolates between edge-preserving and isotropic noise, to ensure that our process converges to the known prior $\mathcal{N}(0, I)$ from which we can start the sampling.
>
> 5. Another crucial component is to ensure smoothness in the transition between edge-preserving and isotropic noise. In the ablation in Section 5.2 of our submission we show that choosing \tau(t) to be linear instead of a steeper function like sigmoid or cosine has a great impact on performance. Additionally, by using constant values for edge sensitivity $\lambda(t)$ instead of a smoother interpolation, the model focuses on a specific granularity of structural content but fails to generate finer details.
>
> In addition, we show how our edge-preserving diffusion process is rather general and fits into both of the two prominent "paradigms" of generative diffusion models. We refer to paragraph 1 of the global response to all reviewers for more details on this.
>
> ## 4. and 5. Confusing equations + ordering in Section 3
> Thank you for pointing out the confusing ordering and use of equations in section 3 of the paper. We agree with this and suggest the following changes:
>
> - Focus our presentation on Eq.(1), since this is the linear equation that forms the template for our linear diffusion process
> - Remove Eq. (2) and (3), as they are not central to our work
> - Engage more early with the work of Perona and Malik that served as inspiration to our work. We would like to introduce the coefficient introduced into Eq. (11) early on, and directly make the link to how we want to integrate it into Eq.(1) to introduce edge preservation into the diffusion process
> - Mention that there are 2 major diffusion paradigms (score-based models and denoising probabilistic models), and how, starting from Eq. (13), our diffusion model fits into both
>
> We are working on an updated version of the paper incorporating these and other changes, which we will upload later during this rebuttal phase.

---

### Official Review · Reviewer_vZw3 · 2024-11-03

**Soundness:** 3
**Presentation:** 2
**Contribution:** 3
**Rating:** 6
**Confidence:** 4

**Summary:**

The paper introduces a novel approach to generative diffusion models by incorporating edge-preserving noise, which is a generalization of denoising diffusion probabilistic models (DDPM). The authors propose an edge-aware noise scheduler that alternates between edge-preserving and isotropic Gaussian noise, aiming to better capture the structural information in data. The model demonstrates faster convergence to the target distribution and improved learning of low-to-mid frequencies, which are crucial for shape and structural representation.

**Strengths:**

1. The paper provides a theoretical foundation for the proposed model, explaining how it generalizes DDPM and improves frequency learning.

2. The model's ability to better preserve structural information could have practical benefits in various image generation and editing applications.

3. The paper presents a creative method for improving diffusion models by introducing edge-preserving noise, addressing a challenge in generative modeling.

4. The method shows substantial improvements in both qualitative and quantitative metrics, particularly in reducing FID scores and enhancing image generation quality.

**Weaknesses:**

1. The paper does not discuss the computational cost of the proposed method compared to existing approaches, which is an important consideration for practical applications. Suggesting comparisons with existing methods on these metrics could provide valuable context. For instance, a table comparing memory usage and speed (inference and training) across baseline models could highlight this model's practical viability more clearly.

2. While the paper demonstrates improvements on a specific dataset, it is unclear how well the model generalizes to other types of models or tasks. Testing on tasks that emphasize diverse structural features, such as denoising or inpainting, could provide insight into how well the model maintains performance across broader applications.

3. The current evaluation uses limited metrics to assess the effectiveness of the proposed method. Including newer metrics like CLIPScore, etc. could provide a more comprehensive view of performance. Specifically, CLIPScore could offer insights into the model's ability to generate semantically consistent images. Given the model's edge-preserving focus, metrics that capture detail preservation and structure alignment might also be beneficial.

4. While the paper compares the proposed method with existing baselines, it could benefit from a more detailed comparative analysis with other state-of-the-art methods, especially those that also aim to preserve structural information in images.

5. The performance of the proposed model may be highly sensitive to the choice of hyperparameters, such as the transition function and edge sensitivity. The paper could provide **more insight** into how these parameters affect the model's performance and **how to select** them effectively.

**Questions:**

None

---

> ### Author Response · Authors · 2024-11-22
> **Response to reviewer vZw3**
>
> Dear reviewer, thank you for your valuable feedback and suggestions. Besides this response directed to you, we have also prepared a global response to all reviewers, discussing common concerns/questions and providing further details.
>
> ## 1. The paper does not discuss the computational cost and memory usage of the proposed method.
> As discussed in paragraph 4 of the global response, our approach adds no significant computation/memory overhead compared to the baseline isotropic models.
>
> ## 2. It is unclear how well the model generalizes to other types of models or tasks
>  We additionally integrate our method into an inpainting application. RePaint (Lugmayr et al. 2022) [2] is a state-of-the-art pixel-space inpainting algorithm. We made a comparative analysis between RePaint and "Edge-preserving RePaint" by performing an inpainting task over 100 images for multiple datasets.  We will also add visual results for this task to the revised version of the paper, which we will upload later in this rebuttal. Note that lower FID is better, and higher CLIPScore is better. We find that our model closely matches the performance of RePaint.
>
> | FID/CLIPScore - Inpainting task  | Ours          | RePaint       |
> |----------------------------------|---------------|---------------|
> | AFHQ-Cat($128^2$)                | 20.50 / 97.91 | 19.77 / 98.21 |
> | CelebA($128^2$)                  | 49.12 / 91.58 | 44.95 / 93.64 |
>
> We would also like to note that, since it was a question of multiple reviewers whether our model is easy to adopt in other settings/frameworks, we have dedicated a paragraph (paragraph 1) in the global response to all reviewers on this concern. In particular, we show how our edge-preserving diffusion is rather general and fits in the two major diffusion model paradigms.
>
> ## 3. Including newer metrics like CLIPScore, etc. could provide a more comprehensive view of performance.
> The table below provides an additional comparison for our shape-guided generative task (cf. SDEdit, Meng et al., 2021) evaluated using the CLIPScore metric. Our method consistently outperforms the baselines on this metric, indicating that the generated images are more semantically aligned with the ground-truths (the original images used to generate the stroke paintings). In the submission, we showed several examples (e.g. Figure 5) where our model solves visual artifacts that are apparent with other baselines, which can improve the semantical meaning of the generated image.
>
> | CLIP score for shape-guided generative task (SDEdit) | Ours      | DDPM  | Simple Diffusion |
> |------------------------------------------------------|-----------|-------|------------------|
> | AFHQ-Cat($128^2$)                                    | **88.97** | 88.78 | 88.23            |
> | CelebA($128^2$)                                      | **61.15** | 61.02 | 60.72            |
> | LSUN-Church($128^2$)                                 | **64.32** | 62.57 | 62.13            |
>
> ## 4. Additional comparative analysis with newer state-of-the-art methods would be helpful, especially those that aim to preserve structural information in the images.
> We do provide additional results for the state-of-the-art model Simple Diffusion (Hoogeboom et al. 2023), which can be found in paragraph 3 of the global discussion. Currently, we are not aware of any other works that are considering structure-aware noise for generative diffusion models in a way that is similar to us. A work with a similar title is "Structure Preserving Diffusion Models" (Lu et al. 2024), however, the focus of this work is very application-specific ; their model aims to be invariant w.r.t. transformations such as translations and rotations (i.e. when $\mathbf{x}_T$ is transformed, $\mathbf{x}_0$ should undergo the exact same transformation). This can be crucial for some applications, such as medical imaging. Instead, our work starts from the observation that diffusion models are denoisers, and aims to make their denoising capabilities more aware of the underlying content by investigating the impact of edge-preserving noise.
>
> ## 5. Please provide more insight into how the parameters affect the model's performance, and how to select them effectively
> In our ablation study (Section 5.2), we discuss and aim to visually show how the hyperparameters impact the models behaviour and performance. Three important parameters significantly affect the model's performance:
>
> - Edge sensitivity $\lambda(t)$
> - Transition function $\tau(t)$
> - Transition point $t_{\phi}$
>
> The major performance improvements come from making $\lambda(t)$ (linearly) time-varying and using a linear $\tau(t)$ instead of a steeper function like a sigmoid. Additionally, the transition point ($t_{\phi}$) significantly impacts the level of detail and sharpness in the generated samples. In the submission, we recommend the point of transition should be kept at $t=\frac{T}{2}$, as this provides an effective balance between sharpness and high-frequency details.

---

> > ### Comment · Reviewer_vZw3 · 2024-11-26
> >
> > Thanks to the author for the detailed rebuttal. After reading the author's rebuttal, I think the manuscript is still valuable. At present, this version of the paper can be given a score of 6. I hope that the subsequent versions can be revised in combination with the rebuttal.

---

### Author Response · Authors · 2024-11-22
**Global response to all reviewers (part 1)**

Dear reviewers, thank you for your valuable feedback and suggestions. This global comment addresses shared concerns and summarizes additional results/comparisons. We are preparing an updated paper with color-coded changes for clarity, to be uploaded later in the rebuttal phase.

## 1. Can our edge-preserving model be made more abstract, so it can be used in the context of other diffusion frameworks? Is it general enough?
Within the diffusion community there are 2 major "paradigms": score-based models and denoising probabilistic models. We show how our diffusion process is rather general and fits in both paradigms, theoretically making it compatible with existing applications located within these paradigms.

Everything starts from the linear equation Eq. (1) in our submission. To make our process edge-preserving, we choose a $\sigma_t$ that suppresses noise, according to the coefficient $\frac{u}{(1-\tau(t)) \sqrt{1 + \frac{||\mathbf{x}_0||}{\lambda(t)}} + \tau(t)}$. For $\gamma_t$ and the numerator $u$, we remained consistent with DDPM's signal and noise coefficients, because the goal of our paper is to measure the mere impact of edge-preserving (non-isotropic) noise on the performance of a generative diffusion model. But nothing prevents us from choosing other values for $\gamma_t$ and $u$.

Given this linear equation, we can choose a paradigm:

**In our paper we follow the denoising probabilistic paradigm:**
Because our diffusion process is linear, we end up with a well-studied Gaussian process with known posteriors, introduced in Section 3 of the paper. By updating the backward posteriors according to our edge-preserving process (Eq. (14) - (16) in our submission) we can generate samples by simulating the backward Gaussian process.

**But, our process also fits in the score-based modelling paradigm:**
We can rewrite our Eq. (13) as the SDE: $d\mathbf{y}_t = \hat{b}(t)dt + \hat{\varsigma} d \mathbf{w}_t$ , where $\hat{b}(t) = \sqrt{\bar{\alpha}_t}$ and $\hat{\varsigma}(t) = \frac{\sqrt{1 - \bar{\alpha}_t}}{(1 - \tau(t))\sqrt{1 + \frac{||\nabla \mathbf{x}_0||}{\lambda(t)}} + \tau(t)}$. We will provide a more detailed derivation and discussion in Appendix B of the revised version of the paper

Briefly said, our model fits in both general diffusion paradigms, and still leaves possibility for many different variants of edge-preserving  processes (e.g. with different $\gamma_t$'s, different numerators of $\sigma_t$). We will be more explicit about this in the revised version which we will submit later in this rebuttal phase. Hopefully this addresses the concern of reviewer mUvY and reviewer WK9j about whether our formulation is general enough and/or tied to existing models.

## 2. How is the method different from DDPM? There seem to be similarities, which might introduce dependencies on DDPM.
As discussed in the paragraph above, our diffusion process fits into (at least) two general widely-adopted paradigms of generative diffusion models. We clarify that our method is not tied to DDPM. While we follow the same paradigm, keep consistent with their signal coefficient $\gamma_t$, and include their $\sigma_t$ in our $\mathbf{\sigma}_t$, this was to solely study the effect of non-isotropic, content-aware noise on a well-studied isotropic diffusion model. That is, we wanted to keep as many parameters consistent as possible, and only measure the impact of edge-preserving noise. Despite that, an infinite amount of choices of the parameters $\gamma_t$ and $\sigma_t$ from Eq. (1) are possible, and nothing prevents us from making completely different choices. Our edge-preserving diffusion process is a process living in the same paradigm, but independent from DDPM.

---

### Author Response · Authors · 2024-11-22
**Global response to all reviewers (part 2)**

## 3. Additional quantitative comparisons with Simple Diffusion (Hoogeboom et al. 2023)
Below we provide comparisons on shape-guided and unconditional image generation with an additional state of the art isotropic baseline, Simple Diffusion (Hoogeboom et al. 2023) [6]. Unfortunately, there is no official implementation made available by the authors, therefore we resorted to this [open-source implementation of Simple Diffusion](https://github.com/faverogian/simpleDiffusion). Note that this implementation does not consider the architectural changes proposed in the paper. Consistent with the results in our submission, our edge-preserving model consistently outperforms Simple Diffusion on the shape-guided generative task based on SDEdit (Meng et al. 2021) [1].

| FID/CLIP score - Shape-guided image generation (SDEdit (Meng et al. 2021))| Ours                  | Simple Diffusion  |
|---------------------------------------------------------------------------|-----------------------|-------------------|
| AFHQ-Cat($128^2$)                                                         | **23.50** / **88.97** | 34.26 / 88.23     |
| CelebA($128^2$)                                                           | **39.08** / **61.15** | 46.43  / 60.72    |
| LSUN-Church($128^2$)                                                      | 56.14 / **64.32**     | **45.93** / 62.13 |

To isolate the impact of edge-preserving noise, it’s fairer to compare the original Simple Diffusion with an edge-preserving version. While we’re still working on this integration due to time constraints, we provide results based on our original setup presented in the submission. Although we get outperformed in unconditional image generation, we are optimistic that incorporating edge-preserving noise into Simple Diffusion will enhance its performance, as we saw with DDPM. We contend that these are reasonable expectations, as both our method and theirs align with the probabilistic framework of known Gaussian processes outlined in the previous paragraph.

| FID - Unconditional image generation             | Ours     | Simple Diffusion |
|--------------------------------------------------|-----------|------------------|
| AFHQ-Cat($128^2$)                                | **13.06** |     15.66        |
| CelebA($128^2$)                                  | 26.17     |     **19.28**    |
| LSUN-Church($128^2$)                             | 23.17     |     **17.87**    |

**Why are our reported FIDs in the double-digits, which seems higher than what state-of-the-art baselines report?**
We trained all baselines and our model for the same number of epochs within a 48-hour, 2x NVIDIA Tesla A40 GPU budget, reflecting real-world training constraints. Using a simple U-Net, our goal was to evaluate the impact of edge-preserving noise, not advanced architectures. With more sophisticated architectures like Simple Diffusion's, we expect FID scores to improve. To motivate the impact of training budget and model architecture, we ran an [open-source implementation of Simple Diffusion](https://github.com/faverogian/simpleDiffusion) with our simple U-Net and found similar double-digit FID scores.

---

### Author Response · Authors · 2024-11-22
**Global response to all reviewers (part 3)**

## 4. Timings for inference and training
Our measurements are based on data resolution (128x128) and a batch size of 64. Note that BNDM and Flow Matching make use of less inference steps (T=250 vs. T=500 for Ours, DDPM and Simple Diffusion), and therefore are expected to be faster for inference. Given that the official implementations of Simple Diffusion (Hoogeboom et al. 2023) and Flow Matching (Lipman et al. 2023) are unavailable, we used [simpleDiffusion](https://github.com/faverogian/simpleDiffusion) and [torchcfm](https://github.com/atong01/conditional-flow-matching) respectively. Our setup consisted of 2 NVIDIA Quadro RTX 8000 GPUS. We see that timings and memory usage of Ours is very similar to DDPM and Simple Diffusion, suggesting that the Sobel filter we apply to approximate $||\nabla \mathbf{x}_0||$ brings minimal overhead. We will add these measurements to the revised version of the paper which we will upload later during the rebuttal phase.


Training:
|                              | Ours | DDPM | BNDM | Flow Matching (Lipman et al. 2022) | Simple Diffusion (Hoogeboom et al. 2023) |
|------------------------------|------|------|------|------------------------------------|------------------------------------------|
| **Time/training iteration in s** | 1.12 | 1.11 | 0.74 | 2.74                               | 1.48                                     |


 Inference:
|                                 | Ours  | DDPM  | BNDM | Flow Matching (Lipman et al. 2022) | Simple Diffusion (Hoogeboom et al. 2023) |
|---------------------------------|-------|-------|------|------------------------------------|------------------------------------------|
| **Inference time in s for 1 batch** | 301.5 | 277.5 | 77.2 | 84.78                              | 290.7                                    |

|                          | Ours | DDPM | BNDM | Flow Matching (Lipman et al. 2022) | Simple Diffusion (Hoogeboom et al. 2023) |
|--------------------------|------|------|------|------------------------------------|------------------------------------------|
| **Total memory usage in GB** | 9.16 | 9.16 | 10.3 | 22.18                              | 10.42                                    |

## Additional comparisons on shape-guided generative task with CLIPScore metric
The table below provides an additional comparison for our shape-guided generative task (cf. SDEdit, Meng et al., 2021) evaluated using the CLIPScore metric. Our method consistently outperforms the baselines on this metric, indicating that the generated images are more semantically aligned with the ground-truths (the original images used to generate the stroke paintings). In the submission, we showed several examples (e.g. Figure 5) where our model solves visual artifacts that are apparent with other baselines, which can improve the semantical meaning of the generated image.

| CLIP score for shape-guided generative task (SDEdit) | Ours      | DDPM  | Simple Diffusion |
|------------------------------------------------------|-----------|-------|------------------|
| AFHQ-Cat($128^2$)                                    | **88.97** | 88.78 | 88.23            |
| CelebA($128^2$)                                      | **61.15** | 61.02 | 60.72            |
| LSUN-Church($128^2$)                                 | **64.32** | 62.57 | 62.13            |

References:
[1]: [SDEdit: Guided Image Synthesis and Editing with Stochastic Differential Equations, Meng et al. 2021](https://arxiv.org/abs/2108.01073)
[2]: [RePaint: Inpainting using Denoising Diffusion Probabilistic Models, Lugmayr et al. 2022](https://arxiv.org/abs/2201.09865)
[3]:[Generative Modelling With Inverse Heat Dissipation, Rissanen et al. 2023](https://arxiv.org/pdf/2206.13397)
[4]:[Generative Modeling by Estimating Gradients of the Data Distribution
, Song et al. 2019](https://arxiv.org/abs/1907.05600)
[5]:[Score-Based Generative Modeling through Stochastic Differential Equations](https://arxiv.org/abs/2011.13456)
[6]: [Simple Diffusion: End-to-end diffusion for high resolution images, Hoogeboom et al. 2023](https://arxiv.org/abs/2301.11093)
[7]: [Flow Matching for Generative Modeling
, Lipman et al. 2023](https://arxiv.org/abs/2210.02747)
[8]: [Denoising Diffusion Probabilistic Models: Ho et al. 2020](https://arxiv.org/abs/2006.11239)

---

### Author Response · Authors · 2024-11-27
**Revision submission**

Dear reviewers,

Thank you all for your constructive feedback in helping us improve the paper. While we are still refining the paper, we wanted to share the current version to illustrate how we plan to incorporate the reviewers' suggestions. We would greatly appreciate any additional feedback to further enhance the revised version.

We have uploaded the revised version, incorporating the following changes (highlighted in blue):

- We have updated the Background section (following suggestions from reviewer mUvY) by introducing discussions on how our edge-preserving model can adapt to other generative paradigms like SDE or score-based generative models (details in Appendix A). This would improve the readability and make a smoother transition towards our formulating based no probabilistic models.
- Added theoretical formulation regarding the relation to Flow Matching (Lipman et al. 2022) in Appendix B, as well as a comparison to Flow Matching in Table 2 (following suggestions from the reviewer WK9j).
- Following WK9j concerns, we explain how negative log-likelihood can be approximated with our formulation in Appendix D.
- We have rewritten equation 11 and the corresponding text for Section 4.1 (following suggestions from the reviewer mUvY).
- Timings and memory consumptions are added in Table 6, showing our method brings minimal overhead compared to DDPM (following suggestions from the reviewer Q9qP).
- We have added additional comparison with Simple Diffusion in Table 4, 5 (following suggestions from the reviewers Q9qP, WK9j).
- Table 7 and 8 are added to perform realism and diversity analysis (following suggestions from the reviewer mUvY).
- We also add new metric comparison in Table 10 with the CLIP score for shape guided image generation (SDEdit) (following suggestions from the reviewers WK9j, Q9qP, vZw3).
- In Table 9/Figure 14, we incorporate our Edge-preserving noise into RePaint (Lugmayr et al., 2022) for image inpainting with visual comparisons (following suggestions from the reviewer vZw3).

Thank you,
The Authors

---

### Meta-Review · Area_Chair_xD3Q · 2024-12-16

**Metareview:**

This paper introduces an edge-preserving diffusion model for image generation. It proposes an edge-aware noise scheduler that varies between edge-preserving and isotropic Gaussian noise, aiming to better capture structural information in images. The authors claim their model converges faster, better learns low-to-mid frequencies crucial for shape representation, and outperforms state-of-the-art baselines in unconditional image generation and tasks guided by a shape-based prior.

The paper introduces a novel edge-preserving approach with a theoretical foundation that generalizes DDPM. It has potential for image generation and editing applications. The authors present both qualitative and quantitative improvements, including a reduction in FID scores. On the negative side, some reviewers mention that the paper's novelty is limited, as it primarily applies existing theory. Additionally, its performance is weak compared to current state-of-the-art methods, and it lacks comparisons with relevant work. The evaluation metrics used are limited, and there is a lack of discussion on generalization to other tasks.

The paper presents an interesting idea, but it is not ready for publication in its current form. The reviewers raise valid concerns regarding the novelty, performance, and evaluation of the proposed method. While the authors have partially addressed some concerns during the rebuttal phase, major issues remain. Specifically, the limited novelty, weak performance, and lack of comprehensive evaluation prevent this paper from meeting the ICLR acceptance threshold.

**Additional Comments On Reviewer Discussion:**

During the rebuttal period, reviewers raised several concerns, including the need for additional comparisons with state-of-the-art methods, clarification on computational cost and generalization to other tasks, and further analysis of hyperparameter sensitivity. The authors responded to these concerns by providing additional experiments and clarifications. However, some reviewers remained unconvinced, particularly due to the weak FIDs and the fact that vanilla flow matching outperformed the proposed method.
For the final decision the following points were reaised:

- Novelty and potential impact: The core idea of incorporating edge-preserving noise is novel and has the potential to improve diffusion models.
- Evaluation: The initial evaluation was limited, and while the authors provided additional results during the rebuttal, concerns about the completeness and rigor of the evaluation remained.
- Reviewer feedback: Some reviewers remained unconvinced after the rebuttal, which raised further concerns about the significance of the contribution.

---

### Decision · Program_Chairs · 2025-01-22

Reject